# A variety of soliton solutions of time M-fractional: Non-linear models via a unified technique

Md. Mamunur Roshid[1,2]☯*, M. M. Rahman[1]☯, Harun-Or Roshid[3], Md. Habibul Bashar[4]

**1** Bangladesh University of Engineering and Technology (BUET), Dhaka, Bangladesh, **2** Department of Mathematics, Hamdard University Bangladesh (HUB), Munshiganj, Bangladesh, **3** Department of Mathematics, Pabna University of Science and Technology (PUST), Pabna, Bangladesh, **4** Department of Mathematics, European University of Bangladesh, Dhaka, Bangladesh

☯ These authors contributed equally to this work.
* mamunmath307@gmail.com, mamunmath307@gmail.com

**Data Availability Statement:** All relevant data are within the manuscript and its Supporting Information files.

**Funding:** The authors received no specific funding for this work. The funders had no role in study

## Abstract

This work explores diverse novel soliton solutions of two fractional nonlinear models, namely the truncated time M-fractional Chafee-Infante (tM-fCI) and truncated time M-fractional Landau-Ginzburg-Higgs (tM-fLGH) models. The several soliton waves of time M-fractional Chafee-Infante model describe the stability of waves in a dispersive fashion, homogeneous medium and gas diffusion, and the solitary waves of time M-fractional Landau-Ginzburg-Higgs model are used to characterize the drift cyclotron movement for coherent ion-cyclotrons in a geometrically chaotic plasma. A confirmed unified technique exploits soliton solutions of considered fractional models. Under the conditions of the constraint, fruitful solutions are gained and verified with the use of the symbolic software Maple 18. Keeping special values of the constraint, this inquisition achieved kink shape, the collision of kink type and lump wave, the collision of lump and bell type, periodic lump wave, bell shape, some periodic soliton waves for time M-fractional Chafee-Infante and periodic lump, and some diverse periodic and solitary waves for time M-fractional Landau-Ginzburg-Higgs model successfully. The required solutions in this work have many constructive descriptions, and corporal behaviors have been incorporated through some abundant 3D figures with density plots. We compare the m-fractional derivative with the beta fractional derivative and the classical form of these models in two-dimensional plots. Comparisons with others' results are given likewise.

## 1. Introduction

Nonlinear phenomena have been effective in diverse research fields. Different nonlinear phenomena occurring in the real world can be conveyed by way of NPDEs, and their real properties are thought of through solitonic solutions observed in several fields such as nonlinear science and engineering [1,2], bio-science [3], hydrodynamics [4], and plasma physics [5,6]. Many researchers investigated solitons and allied them with features of solitary wave solutions such as mono-pulse water movement, which depicts the foremost soliton attained by Feng and Hou [7]. Diverse solitonics are found by Liu [8], Wazwaz [9], Rustam, Saha, and Chatterjee [10], Abdelsalam [11], and

design, data collection and analysis, decision to publish, or preparation of the manuscript.

**Competing interests:** The authors have declared that no competing interests exist.

Roshid et al. [12]. With the progress of soliton theory, there are various effective ways to pursue solitary wave solutions that have been anticipated and advanced, for example, One-Step One-Hybrid Block Method [13], hybrid technique of quintic Hermite splines and weighted finite difference [14], Hirota bilinear [15], generalized Kudryshov [16], multiple exp-function [17], extended tanh [18], double G′/G,1/G-expansion method [19], GERF method [20], NANN method [21], NSE, modified Khater's technique [22], and functions scheme [23], MSE [24], new exponential expansion [25], and singular manifold and group transformation [26] methods, etc.

Fractional derivatives have been widely applied in the analysis of nonlinear evolution equations. It can be implemented to investigate the execution of solutions in various kinds of nonlinear systems. By introducing fractional derivatives, it is possible to express the nonlinear evolution equation in a more general way, which allows for the consideration of nonlinear effects. Fractional derivatives are beneficial in nonlinear models because it can provide more accurate results compared to traditional derivatives, and it hopefully be utilized to analyze solutions with a diverse range of initial conditions. Furthermore, the use of fractional derivatives can reduce the need for tedious parameter changes and simplify the computational process. Recently, many researchers have developed different types of fractional derivatives and applied them to diverse nonlinear systems. M-fractional derivative is used to the Paraxial Wave model [27] and Klein-Gordon model [28], conformable fractional derivative is applied to the date-Jimbo-Kashiwara-Miwa equation [29] and variant Boussinesq equation [30], Caputo fractional derivative is applied to the prey-predator model [31] and Korteweg-de Vries (KdV) models [32], Riemann-Liouville fractional integral is used to the Kudryashov-Sinelshchikov equation [33] and relaxation-oscillation equation [34], fractional beta derivatives is used to Cubic Nonlinear Schrödinger Equation [35] and heat equation [36], etc.

In this paper, we investigate novel truncated M-fractional derivatives via a unified method for the two nonlinear time M-fractional evolution models, such as the Chafee-Infante (tM-fCI) and Landau-Ginburg-Higgs (tM-fLGH) models. There are several methods used for these models in classical differential form [37–43], but still, the truncated M-fractional derivative on the models are not used. This is the first time of exploration of the Chafee-Infante (tM-fCI) and Landau-Ginburg-Higgs (tM-fLGH) models with such fractional form. The (1+1)-dimensional tM-fCI model equation can be presented as follows:

$$D_{M,t}^{\lambda,d}\tau - \tau_{xx} + \alpha(\tau^3 - \tau) = 0, \qquad (1)$$

where $\alpha$ is the coefficient of diffusion, which maintains solitary depiction by modifying the balance between the diffusion term and nonlinear component. The C-I equation has some significant phenomena in the homogeneous medium and gas diffusion [38]. There are many researchers who have investigated the exact solitary wave solutions through diverse methods such as the exp-function method [37], the direct geometric approach [38], the improved Kudryashov method [39], etc.

Besides, a novel nonlinear evolution equation (NLEE) with power law nonlinearity [40] is described as

$$\tau_{tt} + b_1\tau_{xx} + b_2\tau + b_3\tau^p + b_4\tau^{2p-1} = 0,$$

where $b_1,b_2,b_3,b_4$ are free constants. Due to particular values $p = 3,b_4 = 0$, the NLEEs [40] reduce to a special nonlinear structure [41], that states

$$\tau_{tt} + b_1\tau_{xx} + b_2\tau + b_3\tau^3 = 0.$$

Besides this, a typical structure can also be formed from the previous form [41] due to specific parametric constraints $b_1 = -1,b_2 = -a^2,b_3 = b^2$ identified as the LGH model, investigated in [42].

In our view, this model has few gaps to explain accurate real wave nature, and we are going to express more than the above wave, including a truncated M-fractional order derivative on a temporal variable, which can be mentioned as the truncated M-fractional LGH (tM-fLGH) model:

$$D_{M,t}^{2\lambda,d}\tau - \tau_{xx} - a^2\tau + b^2\tau^3 = 0, \tag{2}$$

where the time and space coordinates are t and x, respectively, and the real constants a and b specify the ion-cyclotron wave and electrostatic potential respectively. The LGH Eq (2) was first developed to characterize the drift cyclotron movement for coherent ion-cyclotrons in a geometrically chaotic plasma.

For searching the exact and explicit soliton solutions for the tM-fLGH model by using diverse techniques in mathematical physics such as the homotopy perturbation method [40], the new generalized technique [41], the NMSE technique [42], the IBSEF method [43], and the extended Tanh scheme [44]. In the literature, some typical schemes are used to investigate exact soliton solutions for tM-fCI and tM-fLGH model equations [37–44].

## 2. Preliminary fractional derivative

**Characterization:** let us consider a mapping $\mathfrak{H}:(0,\infty)\rightarrow\mathbb{R}$, the $p$ order Truncated M-derivative of $\mathfrak{H}$ exhibit as:

$$D_{M,t}^{n,d}\mathfrak{H}(t) = \lim_{h\to 0}\frac{\mathfrak{H}(tH_d(ht^{1-n})) - \mathfrak{H}(t)}{h}.; \ d > 0, 0 < p < 1.$$

Here $H_d(z)$ is a one-parameter truncated Mittag-Leffler function that is well-defined as [45,46]:

$$H_d(z) = \sum_{j=0}^{i}\frac{z^j}{\Gamma(dj+1)}.$$

**Mannerisms**:
Assume d>0,0<p<1,m,n∈ℜ and $\mathfrak{H}$,$g$,p−differentiable at a point t>0, then

$$(1) \ \ D_{M,t}^{p,d}(m\mathfrak{H}(t) + n \quad (t)) = pD_{M,t}^{p,d}\mathfrak{H}(t) + qD_{M,t}^{p,d} \quad (t).$$

$$(2) \ \ D_{M,t}^{p,d}(\mathfrak{H}(t) \quad (t)) = \mathfrak{H}(t)D_{M,t}^{p,d} \quad (t) + \quad (t)D_{M,t}^{p,d}\mathfrak{H}(t).$$

$$(3) \ \ D_{M,t}^{p,d}\left(\frac{\mathfrak{H}(t)}{(t)}\right) = \frac{\mathfrak{H}(t)D_{M,t}^{p,d} \quad (t) + \quad (t)D_{M,t}^{p,d}\mathfrak{H}(t)}{(t)^2}.$$

$$(4) \ \ D_{M,t}^{p,d}\mathfrak{H}(t) = 0 \text{ where } \mathfrak{H}(t) = a.$$

$$(5) \ \ D_{M,t}^{p,d}\mathfrak{H}(t) = \frac{t^{1-p}}{\Gamma(d+1)}\frac{d\mathfrak{H}(t)}{dt}.$$

## 3. Unified method

In this section, to explore the unified scheme [27,47], we consider a higher dimensional PDEs:

$$G[\mathfrak{H}] = G(\mathfrak{H}, \mathfrak{H}_{xx}, D_{M,t}^{\lambda,d}\mathfrak{H}, D_{M,t}^{\lambda,d}(\mathfrak{H}_{xx})\ldots\ldots\ldots), \tag{3}$$

here $\mathfrak{H} = \mathfrak{H}(x,t)$.

To solve the Eq (3), the essential steps of the unified scheme are below:

**Step-01**: At first, incorporating the directional wave variable

$\mathfrak{H}(x,t) = \mathfrak{H}(\eta); \eta = kx - \omega\frac{\Gamma(d+1)}{\lambda}t^{\lambda}$, the PDE (3) becomes an ordinary differential equation. Then we get,

$$R[\mathfrak{H}] = R(f, k^2\mathfrak{H}_{\eta\eta}, \omega\mathfrak{H}_{\eta}, k^2\omega\mathfrak{H}_{\eta\eta\eta} \ldots\ldots\ldots). \tag{4}$$

**Step-02**: Consider the solution of Eq (4) is:

$$\mathfrak{H}(\eta) = a_0 + \sum_{s=1}^{N}[\mu_p\wp(\eta)^s + \vartheta_p\wp(\eta)^{-s}], \tag{5}$$

Where $\mu_S, \vartheta_S$ (p = 0,1,...,N) are unknown constants. The trail solution Eq (5) satisfies the Riccati differential equation:

$$\wp(\eta)' = (\wp(\eta))^2 + \gamma. \tag{6}$$

To find the balance number *N*, we use the following formula:

$$\frac{d^p\mathfrak{H}}{d\eta^p} = N + p, \mathfrak{H}^l\frac{d^p\mathfrak{H}}{d\eta^p} = lN + N + p, \left(\frac{d^p\mathfrak{H}}{d\eta^p}\right)^S = S(N + p). \tag{7}$$

**Step-03**: Injecting different form of Eq (5) into Eq (4), one gets a sequence of $\wp^{\pm h}, h \in \mathbb{R}$. The set of equations is formed if the coefficient of $\wp^{\pm h}$ is thought to be zero. By using Maple 18 to solve the obtained system, the expected solution sets are attained.

Under the conditions on the parameter of Eq (6) the following solution are gotten:

For γ<0,

$$\wp(\eta) = \begin{cases} \dfrac{\sqrt{-(h^2 + \ell^2)\gamma} - h\sqrt{-\gamma}\cosh(2\sqrt{-\gamma}(\eta + p))}{h\sinh(2\sqrt{-\gamma}(\eta + p)) + \ell}, \\[2ex] \dfrac{-\sqrt{-(h^2 + \ell^2)\gamma} - h\sqrt{-\gamma}\cosh(2\sqrt{-\gamma}(\eta + p))}{h\sinh(2\sqrt{-\gamma}(\eta + p)) + \ell}, \\[2ex] \sqrt{-\gamma} + \dfrac{-2h\sqrt{-\gamma}}{h + \cosh(2\sqrt{-\gamma}(\eta + p)) - \sinh(2\sqrt{-\gamma}(\eta + p))}, \\[2ex] -\sqrt{-\gamma} + \dfrac{2h\sqrt{-\gamma}}{h + \cosh(2\sqrt{-\gamma}(\eta + p)) - \sinh(2\sqrt{-\gamma}(\eta + p))}, \end{cases}.$$

For γ>0,

$$\wp(\eta) = \begin{cases} \dfrac{\sqrt{(h^2 - \ell^2)\gamma} - h\sqrt{\gamma}\cos(2\sqrt{\gamma}(\eta + p))}{h\sin(2\sqrt{\gamma}(\eta + p)) + \ell}, \\[2ex] \dfrac{-\sqrt{(h^2 - \ell^2)\gamma} - h\sqrt{\gamma}\cos(2\sqrt{\gamma}(\eta + p))}{h\sin(2\sqrt{\gamma}(\eta + p)) + \ell}, \\[2ex] i\sqrt{\gamma} + \dfrac{-2hi\sqrt{\gamma}}{h + \cos(2\sqrt{\gamma}(\eta + p)) - i\sin(2\sqrt{\gamma}(\eta + p))}, \\[2ex] -i\sqrt{\gamma} + \dfrac{2hi\sqrt{\gamma}}{h + \cos(2\sqrt{\gamma}(\eta + p)) + i\sin(2\sqrt{\gamma}(\eta + p))}, \end{cases}.$$

For $\gamma = 0$,

$$\wp(\eta) = -\frac{1}{\eta + p}.$$

N.B: For the beta fractional derivative, we used the beta time-fractional derivative is:

$$\eta = kx - \omega \frac{\left(\frac{1}{\Gamma(d)} + t\right)^{\lambda}}{\lambda}.$$

## 4. Application of UN scheme

In this section, we operate the fractional unified scheme to generate some novel soliton solution for **the** tM-fCI and the tM-fLGH models.

### 4.1 Time M-fractional Chafee-Infante (tM-fCI) equation

We start the time M-fractional Chafee-Infante equation in the following form:

$$D_{M,t}^{\lambda,d}\tau - \tau_{xx} - \alpha(\tau - \tau^3) = 0. \tag{8}$$

Using truncated M-fractional derivative, and the transformation variable, one ahead to $D_{M,t}^{\lambda,d}\tau(x,t) = -\omega\frac{d\tau(\eta)}{d\eta}$ and $\eta = kx - \omega\frac{\Gamma(d+1)}{\lambda}t^{\lambda}$. Then the Eq (8) becomes,

$$-\omega\frac{d\tau}{d\eta} + k^2\frac{d^2\tau}{d\eta^2} + \alpha(\tau - \tau^3) = 0. \tag{9}$$

According to Eq (7), the solution of Eq (9) is

$$\tau = \mu_0 + \mu_1\varphi(\eta) + \frac{\mu_2}{\varphi(\eta)}. \tag{10}$$

Here $\mu_0, \mu_1$ and $\mu_2$ are unfamiliar parameters. Eqs (6) and (10) have been placed into Eq (9), and in agreement with step-03, we now possess a system of equations.

$$-2k^2\mu_1 + \alpha\mu_1^3 = 0.$$

$$3\alpha\mu_0\mu_1^2 - \omega\mu_1 = 0.$$

$$-2\gamma k^2\mu_1 + 3\alpha\mu_1\mu_0^2 + 3\alpha\mu_2\mu_1^2 - \alpha\mu_1 = 0.$$

$$-\omega\gamma\mu_1 + \alpha\mu_0^3 + 6\alpha\mu_0\mu_1\mu_2 + \omega\mu_2 - \alpha\mu_0 = 0.$$

$$-2\gamma k^2\mu_2 + 3\alpha\mu_2\mu_0^2 + 3\alpha\mu_1\mu_2^2 - \alpha\mu_2 = 0.$$

$$3\alpha\mu_0\mu_1^2 + \gamma\omega\mu_2 = 0.$$

$$-2\gamma^2 k^2\mu_2 + \alpha\mu_2^3 = 0.$$

To get the required solution set, the above system is solved by using software Maple 18.

**Set-01**: If we make use of the parameters $k = \sqrt{\frac{-\alpha}{8\gamma}}, \omega = \frac{3\alpha}{2}\sqrt{\frac{-1}{4\gamma}}, \mu_0 = \frac{1}{2}, \mu_1 = \sqrt{\frac{-1}{4\gamma}}, \mu_2 = 0$ in Eq (10), then we obtain the following solutions.

For γ<0, the hyperbolic solutions are

$$u_{1,1} = \frac{1}{2} + \sqrt{\frac{-1}{4\gamma}} \frac{\sqrt{-(h^2+\ell^2)\gamma} - h\sqrt{-\gamma}\cosh(2\sqrt{-\gamma}(\eta+p))}{h\sinh(2\sqrt{-\gamma}(\eta+p)) + \ell}, \tag{11}$$

$$u_{1,2} = \frac{1}{2} + \sqrt{\frac{-1}{4\gamma}} \frac{-\sqrt{-(h^2+\ell^2)\gamma} - h\sqrt{-\gamma}\cosh(2\sqrt{-\gamma}(\eta+p))}{h\sinh(2\sqrt{-\gamma}(\eta+p)) + \ell}, \tag{12}$$

$$u_{1,3} = \frac{1}{2} + \sqrt{\frac{-1}{4\gamma}}\left(\sqrt{-\gamma} + \frac{-2h\sqrt{-\gamma}}{h + \cosh(\sqrt{-4\gamma}(\eta+p)) - \sinh(\sqrt{-4\gamma}(\eta+p))}\right), \tag{13}$$

$$u_{1,4} = \frac{1}{2} + \sqrt{\frac{-1}{4\gamma}}\left(-\sqrt{-\gamma} + \frac{2h\sqrt{-\gamma}}{h + \cosh(\sqrt{-4\gamma}(\eta+p)) - \sinh(\sqrt{-4\gamma}(\eta+p))}\right), \tag{14}$$

For γ>0, the trigonometric solutions are

$$u_{1,5} = \frac{1}{2} + \sqrt{\frac{-1}{4\gamma}} \frac{\sqrt{(h^2-\ell^2)\gamma} - h\sqrt{\gamma}\cos(2\sqrt{\gamma}(\eta+p))}{h\sin(2\sqrt{\gamma}(\eta+p)) + \ell}, \tag{15}$$

$$u_{1,6} = \frac{1}{2} + \sqrt{\frac{-1}{4\gamma}} \frac{-\sqrt{(h^2-\ell^2)\gamma} - h\sqrt{\gamma}\cos(2\sqrt{\gamma}(\eta+p))}{h\sin(2\sqrt{\gamma}(\eta+p)) + \ell}, \tag{16}$$

$$u_{1,7} = \frac{1}{2} + \sqrt{\frac{-1}{4\gamma}}\left(i\sqrt{\gamma} + \frac{-2hi\sqrt{\gamma}}{h + \cos(2\sqrt{\gamma}(\eta+p)) - i\sin(2\sqrt{\gamma}(\eta+p))}\right), \tag{17}$$

$$u_{1,8} = \frac{1}{2} + \sqrt{\frac{-1}{4\gamma}}\left(-i\sqrt{\gamma} + \frac{2hi\sqrt{\gamma}}{h + \cos(\sqrt{4\gamma}(\eta+p)) + i\sin(\sqrt{4\gamma}(\eta+p))}\right), \tag{18}$$

here $\eta = x\sqrt{\frac{-\alpha}{8\gamma}} - \frac{3\alpha}{2}\sqrt{\frac{-1}{4\gamma}}\frac{\Gamma(d+1)}{\lambda}t^\lambda$ and h,ℓ,α,γ,p are arbitrary constants.

**Set-02**: If we insert the value of the parameters $k = \sqrt{\frac{-\alpha}{8\gamma}}, \omega = -\frac{3\alpha}{4\gamma}\sqrt{-\gamma}, \mu_0 = \frac{1}{2}, \mu_1 = 0, \mu_2 = \frac{\sqrt{-\gamma}}{2}$ in Eq (10), then we get the specific solutions.

For γ<0, the hyperbolic solutions are

$$u_{2,1} = \frac{1}{2} + \frac{\sqrt{-\gamma}}{2}\frac{h\sinh(2\sqrt{-\gamma}(\eta+p)) + \ell}{\sqrt{-(h^2+\ell^2)\gamma} - h\sqrt{-\gamma}\cosh(2\sqrt{-\gamma}(\eta+p))}, \tag{19}$$

$$u_{2,2} = \frac{1}{2} + \frac{\sqrt{-\gamma}}{2}\frac{h\sinh(2\sqrt{-\gamma}(\eta+p)) + \ell}{-\sqrt{-(h^2+\ell^2)\gamma} - h\sqrt{-\gamma}\cosh(2\sqrt{-\gamma}(\eta+p))} \tag{20}$$

$$u_{2,3} = \frac{1}{2} + \frac{\sqrt{-\gamma}}{2}\frac{1}{\sqrt{-\gamma} + \frac{-2h\sqrt{-\gamma}}{h+\cosh(\sqrt{-4\gamma}(\eta+p))-\sinh(\sqrt{-4\gamma}(\eta+p))}} \tag{21}$$

$$u_{2,4} = \frac{1}{2} + \frac{\sqrt{-\gamma}}{2} \frac{1}{-\sqrt{-\gamma} + \frac{2h\sqrt{-\gamma}}{h+\cosh(\sqrt{-4\gamma}(\eta+p))-\sinh(\sqrt{-4\gamma}(\eta+p))}}, \tag{22}$$

here $\eta = x\sqrt{\frac{-\alpha}{8\gamma}} + \frac{3\alpha}{2}\sqrt{\frac{-1}{4\gamma}}\frac{\Gamma(d+1)}{\lambda}t^{\lambda}$ and $h, \ell, \alpha, \gamma, p$ are arbitrary constants.

For $\gamma > 0$, the trigonometric solutions are

$$u_{2,5} = \frac{1}{2} + \frac{\sqrt{-\gamma}}{2} \frac{1}{\frac{\sqrt{(h^2-\ell^2)\gamma}-h\sqrt{\gamma}\cos(2\sqrt{\gamma}(\eta+p))}{h\sin(2\sqrt{\gamma}(\eta+p))+\ell}}, \tag{23}$$

$$u_{2,6} = \frac{1}{2} + \frac{\sqrt{-\gamma}}{2} \frac{1}{\frac{-\sqrt{(h^2-\ell^2)\gamma}-h\sqrt{\gamma}\cos(2\sqrt{\gamma}(\eta+p))}{h\sin(2\sqrt{\gamma}(\eta+p))+\ell}}, \tag{24}$$

$$u_{2,7} = \frac{1}{2} + \frac{\sqrt{-\gamma}}{2} \frac{1}{i\sqrt{\gamma} + \frac{-2hi\sqrt{\gamma}}{h+\cos(\sqrt{4\gamma}(\eta+p))-i\sin(\sqrt{4\gamma}(\eta+p))}}, \tag{25}$$

$$u_{2,8} = \frac{1}{2} + \frac{\sqrt{-\gamma}}{2} \frac{1}{-i\sqrt{\gamma} + \frac{2hi\sqrt{\gamma}}{h+\cos(\sqrt{4\gamma}(\eta+p))+i\sin(\sqrt{4\gamma}(\eta+p))}}, \tag{26}$$

here $\eta = x\sqrt{\frac{-\alpha}{8\gamma}} + \frac{3\alpha}{2}\sqrt{\frac{-1}{4\gamma}}\frac{\Gamma(d+1)}{\lambda}t^{\lambda}$ and $h, \ell, \alpha, \gamma, p$ are arbitrary constants.

## 4.2. Time M-fractional Landau-Ginburg-Higgs equation (tM-fLGHE)

Let us consider the tM-fLGH model in the following form:

$$D_{M,t}^{2\lambda,d}\tau - \tau_{xx} - a^2\tau + b^2\tau^3 = 0, \tag{27}$$

Using the transformation variable $\tau(x,t) = \tau(\eta)$ and $\eta = kx - \omega\frac{\Gamma(d+1)}{\lambda}t^{\lambda}$ to Eq (27), we get

$$\left(\omega^2 - k^2\right)\frac{d^2\tau}{d\eta^2} - a^2\tau + b^2\tau^3 = 0. \tag{28}$$

Let the trial solution of Eq (27) is:

$$\tau = \mu_0 + \mu_1\varphi(\eta) + \frac{\mu_2}{\varphi(\eta)}, \tag{29}$$

here $\mu_2, \mu_1$ and $\mu_0$ are the unfamiliar. In accordance with steps 02 and 03 the Eqs (6), (28) and (29) provides the succeeding set of equations:

$$\mu_1^3 b^2 - 2k^2\mu_1 + 2\mu_1\omega^2 = 0.$$

$$3b^2\mu_0\mu_1^2 = 0.$$

$$-2\gamma k^2\mu_1 + 3b^2\mu_1\mu_0^2 + 3b^2\mu_2\mu_1^2 + 2\gamma\omega^2\mu_1 - a^2\mu_1 = 0.$$

$$b^2\mu_0^3 + 6b^2\mu_0\mu_1\mu_2 - a^2\mu_0 = 0.$$

$$-2\gamma k^2 \mu_2 + 2\gamma \omega^2 \mu_2 + 3b^2 \mu_2 \mu_0^2 + 3b^2 \mu_1 \mu_2^2 - a^2 \mu_2 = 0.$$

$$3b^2 \mu_0 \mu_1^2 = 0.$$

$$\mu_2^3 b^2 - 2\gamma^2 k^2 \mu_2 + 2\gamma^2 \omega^2 \mu_2 = 0.$$

To get the required solution set, the above system is solved by using software Maple 18.

**Set-01**: Inserting the value of the parameters $\omega = \frac{3}{2}\sqrt{\frac{-1}{4\gamma}}, \mu_0 = 0, \mu_1 = \frac{a}{\sqrt{2}b}\sqrt{\frac{1}{\gamma}}, \mu_2 = \frac{a}{\sqrt{2}b}\sqrt{\gamma}$

in Eq (10) obtain the solutions as:

For $\gamma < 0$, the hyperbolic solutions are attained,

$$u_{1,1} = \frac{a}{\sqrt{2}b}\left(\sqrt{\frac{1}{\gamma}}\frac{\sqrt{-\gamma(h^2+\ell^2)}-h\sqrt{-\gamma}\cosh(2\sqrt{-\gamma}(\eta+p))}{h\sinh(2\sqrt{-\gamma}(\eta+p))+\ell}\right. \\ \left.+\sqrt{\gamma}\frac{h\sinh(\sqrt{-4\gamma}(\eta+p))+\ell}{\sqrt{-\gamma(h^2+\ell^2)}-h\sqrt{-\gamma}\cosh(\sqrt{-4\gamma}(\eta+p))}\right), \tag{30}$$

$$u_{1,2} = \frac{a}{\sqrt{2}b}\left(\sqrt{\frac{1}{\gamma}}\frac{-\sqrt{-\gamma(h^2+\ell^2)}-h\sqrt{-\gamma}\cosh(2\sqrt{-\gamma}(\eta+p))}{h\sinh(2\sqrt{-\gamma}(\eta+p))+\ell}\right. \\ \left.+\sqrt{\gamma}\frac{h\sinh(\sqrt{-4\gamma}(\eta+p))+\ell}{-\sqrt{-\gamma(h^2+\ell^2)}-h\sqrt{-\gamma}\cosh(\sqrt{-4\gamma}(\eta+p))}\right), \tag{31}$$

$$u_{1,3} = \frac{2\sqrt{2}ah(\cosh(2\sqrt{-\gamma}(\eta+p))-\sinh(2\sqrt{-\gamma}(\eta+p)))}{bh((\cosh(2\sqrt{-\gamma}(\eta+p))-\sinh(2\sqrt{-\gamma}(\eta+p)))^2-h^2)}, \tag{32}$$

$$u_{1,4} = \frac{2\sqrt{2}ah(\cosh(\sqrt{-4\gamma}(\eta+p))+\sinh(\sqrt{-4\gamma}(\eta+p)))}{bh((\cosh(\sqrt{-4\gamma}(\eta+p))+\sinh(\sqrt{-4\gamma}(\eta+p)))^2-h^2)}, \tag{33}$$

where $\eta = kx + \frac{3}{2}\sqrt{\frac{-1}{4\gamma}}\frac{\Gamma(d+1)}{\lambda}t^\lambda$ and $h,\ell,a,b,\gamma,p$ are arbitrary constants.

For $\gamma > 0$, the trigonometric solutions are

$$u_{1,5} = \frac{a}{\sqrt{2}b}\left(\sqrt{\frac{1}{\gamma}}\frac{\sqrt{\gamma(h^2-\ell^2)}-h\sqrt{\gamma}\cos(2\sqrt{\gamma}(\eta+p))}{h\sin(\sqrt{4\gamma}(\eta+p))+\ell}\right. \\ \left.+\sqrt{\gamma}\frac{h\sin(\sqrt{4\gamma}(\eta+p))+\ell}{\sqrt{\gamma(h^2-\ell^2)}-h\sqrt{\gamma}\cos(\sqrt{4\gamma}(\eta+p))}\right), \tag{34}$$

$$u_{1,6} = \frac{-a}{\sqrt{2}b}\left(\sqrt{\frac{1}{\gamma}}\frac{-\sqrt{(h^2-\ell^2)\gamma}-h\sqrt{\gamma}\cos(2\sqrt{\gamma}(\eta+p))}{h\sin(2\sqrt{\gamma}(\eta+p))+\ell}\right. \\ \left.+\sqrt{\gamma}\frac{h\sin(2\sqrt{\gamma}(\eta+p))+\ell}{-\sqrt{(h^2-\ell^2)\gamma}-h\sqrt{\gamma}\cos(2\sqrt{\gamma}(\eta+p))}\right), \tag{35}$$

$$u_{1,7} = \frac{2\sqrt{2}ah(h\cos(2\sqrt{\gamma}(\eta+p)) + \sin(2\sqrt{\gamma}(\eta+p)))}{b(2h\cos(2\sqrt{\gamma}(\xi+p))\sin(2\sqrt{\gamma}(\eta+p)) - 2(\cos(2\sqrt{\gamma}(\eta+p)))^2 + 1 + h^2)}, \qquad (36)$$

$$u_{1,8} = \frac{2\sqrt{2}ah(h\cos(2\sqrt{\gamma}(\eta+p)) - \sin(2\sqrt{\gamma}(\eta+p)))}{b(2h\cos(2\sqrt{\gamma}(\xi+p))\sin(2\sqrt{\gamma}(\eta+p)) + 2(\cos(2\sqrt{\gamma}(\eta+p)))^2 - 1 - h^2)}, \qquad (37)$$

where $\eta = kx + \frac{3}{2}\sqrt{\frac{-1}{4\gamma}}\frac{\Gamma(d+1)}{\lambda}t^\lambda$ and h,ℓ,a,b,γ,p are arbitrary constants.

**Set-02**: Substituting the parameters $\omega = \frac{1}{2}\sqrt{\frac{4\gamma k^2 + 2a^2}{\gamma}}, \mu_0 = 0, \mu_1 = \frac{a}{b}\sqrt{\frac{-1}{\gamma}}, \mu_2 = 0$ in Eq (10), we achieve solutions are

For γ<0, the hyperbolic solutions are

$$u_{2,1} = \frac{1}{2} + \frac{a}{b}\sqrt{\frac{-1}{\gamma}}\frac{\sqrt{-(h^2+\ell^2)\gamma} - h\sqrt{-\gamma}\cosh(2\sqrt{-\gamma}(\eta+p))}{h\sinh(2\sqrt{-\gamma}(\eta+p)) + \ell}, \qquad (38)$$

$$u_{2,2} = \frac{1}{2} + \sqrt{\frac{-1}{4\gamma}}\frac{-\sqrt{-\gamma(h^2+\ell^2)} - h\sqrt{-\gamma}\cosh(\sqrt{-4\gamma}(\eta+p))}{h\sinh(\sqrt{-4\gamma}(\eta+p)) + \ell}, \qquad (39)$$

$$u_{2,3} = \frac{1}{2} + \sqrt{\frac{-1}{4\gamma}}\left(\sqrt{-\gamma} + \frac{-2h\sqrt{-\gamma}}{h + \cosh(\sqrt{-4\gamma}(\eta+p)) - \sinh(\sqrt{-4\gamma}(\eta+p))}\right), \qquad (40)$$

$$u_{2,4} = \frac{1}{2} + \sqrt{\frac{-1}{4\gamma}}\left(-\sqrt{-\gamma} + \frac{h\sqrt{-4\gamma}}{h + \cosh(\sqrt{-4\gamma}(\eta+p)) - \sinh(\sqrt{-4\gamma}(\eta+p))}\right), \qquad (41)$$

where $\eta = kx + \frac{1}{2}\sqrt{\frac{4\gamma k^2 + 2a^2}{\gamma}}\frac{\Gamma(d+1)}{\lambda}t^\lambda$ and h,ℓ,a,b,γ,p are arbitrary constants.

For γ>0, the trigonometric solutions are

$$u_{2,5} = \frac{1}{2} + \sqrt{\frac{-1}{4\gamma}}\frac{\sqrt{(h^2-\ell^2)\gamma} - h\sqrt{\gamma}\cos(2\sqrt{\gamma}(\eta+p))}{h\sin(2\sqrt{\gamma}(\eta+p)) + \ell}, \qquad (42)$$

$$u_{2,6} = \frac{1}{2} + \sqrt{\frac{-1}{4\gamma}}\frac{-\sqrt{(h^2-\ell^2)\gamma} - h\sqrt{\gamma}\cos(2\sqrt{\gamma}(\eta+p))}{h\sin(2\sqrt{\gamma}(\eta+p)) + \ell}, \qquad (43)$$

$$u_{2,7} = \frac{1}{2} + \sqrt{\frac{-1}{4\gamma}}\left(i\sqrt{\gamma} + \frac{-2hi\sqrt{\gamma}}{h + \cos(2\sqrt{\gamma}(\eta+p)) - i\sin(2\sqrt{\gamma}(\eta+p))}\right), \qquad (44)$$

$$u_{2,8} = \frac{1}{2} + \sqrt{\frac{-1}{4\gamma}}\left(-i\sqrt{\gamma} + \frac{2hi\sqrt{\gamma}}{h + \cos(2\sqrt{\gamma}(\eta+p)) + i\sin(2\sqrt{\gamma}(\eta+p))}\right), \qquad (45)$$

**4.2.1 Djfdsjkb.**   where $\eta = kx + \frac{1}{2}\sqrt{\frac{4\gamma k^2 + 2a^2}{\gamma}}\frac{\Gamma(d+1)}{\lambda}t^\lambda$ and h,ℓ,a,b,γ,p are arbitrary constants.

## 5. Result and discussion

The unified method has been successfully implemented on the truncated M-fractional differential models to obtain some novel soliton solutions. The novel dynamical solutions have been attained by operating this scheme for two powerful models, the tM-fCI and tM-fLGH models. The parameter of the novel truncated M-fractional derivative has influenced the behavior of attained solutions and also compare the M-fractional derivative with beta fractional derivative and classical form of the differential models. The proposed answers are innovative and have significant implications for illuminating intricate physical nonlinear systems in optical communications.

### 5.1 The time truncated M-fractional Chafee-Infante equation

In this subsector, we graphically demonstrate the impact of fractional parameters on derived soliton solutions of the tM-fCI equation with the three-dimensional graphs and the compare the M-fractional derivative with the beta fractional and classical form of the Chafee-Infante equation in the two-dimensional graphs. The observed solutions express themselves as hyperbolic, rational, and trigonometric function forms with concise descriptions of numerous new types of traveling speeds under varied conditions. The obtained solutions have many significant to describe diverse phenomena in the homogeneous medium and gas diffusion.

For $\gamma < 0$, the unified approach provides hyperbolic function solutions in Eqs (11)–(14). Fig 1 displays from the numerical form of the derived hyperbolic solutions. In Fig 1, the kink-shaped wave solution is achieved for diverse value of the fractional parameter [$\lambda =$ 0.1,0.5,0.9] at $\alpha = d = 0.5 = -\gamma$, h = 5, $\ell = -4$, q=1. The obtained solution is illustrated with three dimensional plots with density. The solutions in Eqs (11) through (14) provided a variety of structures, such as those seen in Figs 2–4.

For $\gamma > 0$, the unified approach provides trigonometric function solutions in Eqs (15)–(18). The numerical values of the parameters in the obtained trigonometric function solution are explored with three dimensional, and density plots. In Fig 2, The impact of $\lambda$ on the solution of interaction between lump and kink wave for $\alpha = 0.167$, d = $0.5 = \gamma$, h = 5, $\ell = -4$, p = 1 with 3-D and corresponding density plots. The feature of lump wave solution of Eq (15) is illustrated in Fig 3, accordance with the influence of the parameter $\lambda$ such that [$\lambda = 0.1,0.3,0.5,0.7,0.9$] at $\gamma =$ 0.1, h = 1, $\alpha = -0.25$, $\ell = 0.5$, d = 0.5, p = 1. In Fig 4, the solution Eq (17) depicted periodic lump wave for $\alpha = -0.167$, d = $0.5$, $\gamma = 0.5$, h = 5, $\ell = -4$, p = 1 with three dimensional and the corresponding density plots.

For $\gamma < 0$, the unified approach provides hyperbolic function solutions, Eqs (19)–(22). In Fig 5, we show the feature of interaction of kink soliton with lump wave via Eq (19) for $\gamma = -0.5, h = -1, \alpha = 1, \ell = 0.2, d = 0.5, p = -1$. The solution of Eq (19) represents soliton solution with the influence of the parameter $\gamma = -0.5, d = .5, h = 0.25, p = -1, \alpha = .5$, illustrates in Fig 6.

For $\gamma > 0$, the unified approach provides trigonometric function solutions in Eqs (23)–(26). The Fig 7 presents the feature of the collision of kink and lump wave solution, Eq (23), due to changes of the parameter as $\lambda$ for $d = 0.0035, \gamma = 0.1, h = 4, \alpha = 0.5, \ell = 0.5, p = 1$. The feature of interaction between lump and bell type soliton solution of Eq (23) shows in Fig 8 due to changes of the parameter for $\alpha = 0.5, d = 1, \gamma = 0.1, h = 4, l = 0.5, p = 1$. In Fig 9, we get the feature of bright bell shape solution of Eq (23) for the parameter $\gamma = -0.5, d = 0.5, h = 4, l = -4, \alpha = 0.5, p = 1$. In Fig 10, we show the feature of dark bell type solution of Eq (23) for the parameter $d = .5 = -\gamma, h = 5, \alpha = 0.5, l = -4, p = 1$.

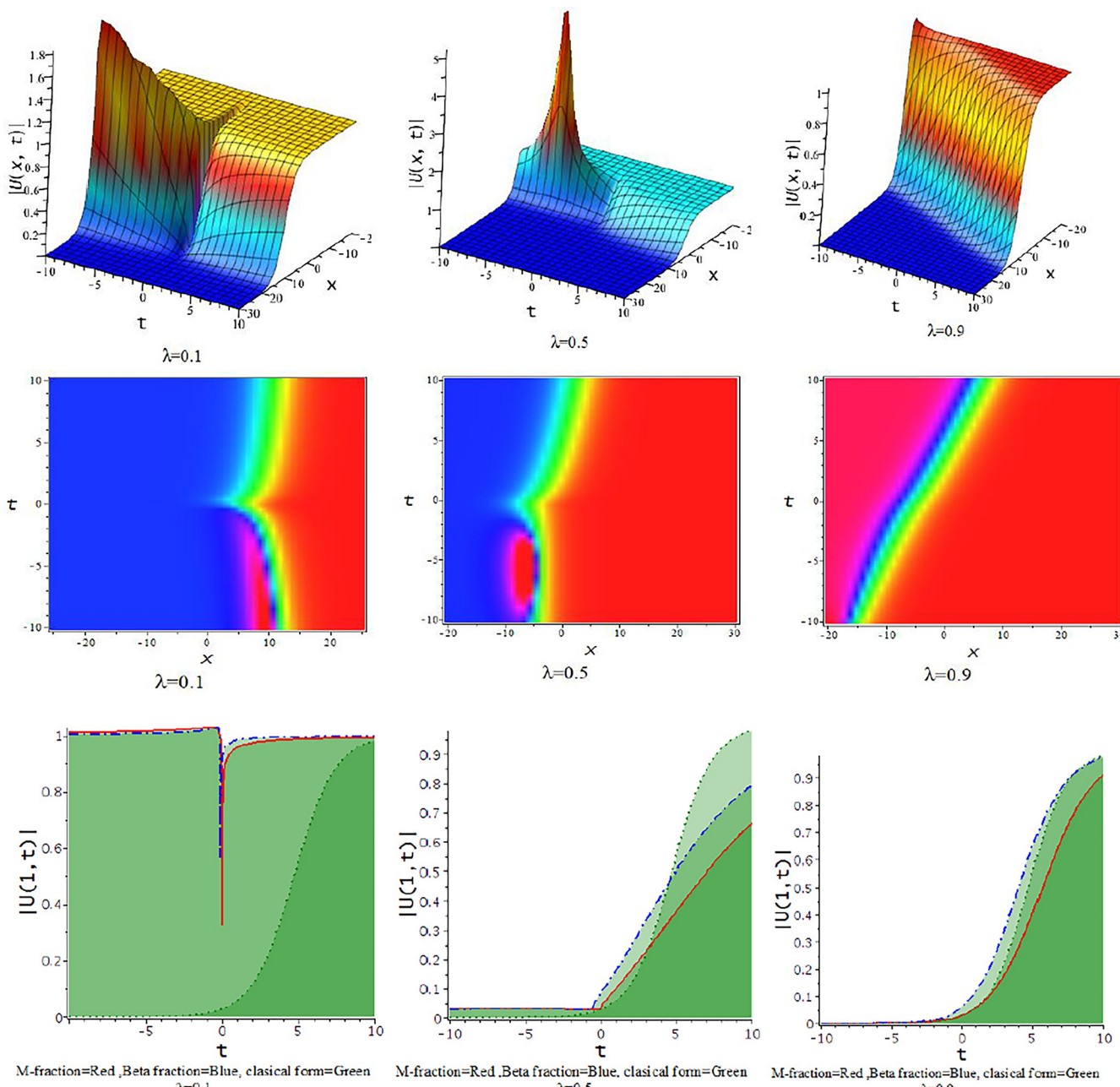

**Fig 1. The feature of kink shape soliton solution of Eq (13) with changes of the parameter as [λ = 0.1,0.5,0.9] at α = d = 0.5 = −γ,h = 5,ℓ = −4, p = 1.**

## 5.2 The time truncated M-fractional Landau-Ginzburg-Higgs equation

In this subdivision, we graphically illustrate the impact of the fractional parameter [λ = 0.3,0.6,0.9] on the derived soliton solution of the tM-fLGH equation and compare the effect of M-fractional derivative with beta fractional derivative and classical form in two-dimensional graph. Fig 11 to 16 depict the dynamic characteristics of the solutions that were obtained from the tM-fLGH equation. The LGH Eq (2) was first developed to characterize the drift cyclotron movement for coherent ion-cyclotrons in a geometrically chaotic plasma. The obtained solutions have many significant to describe the theoretical framework used in particle physics and

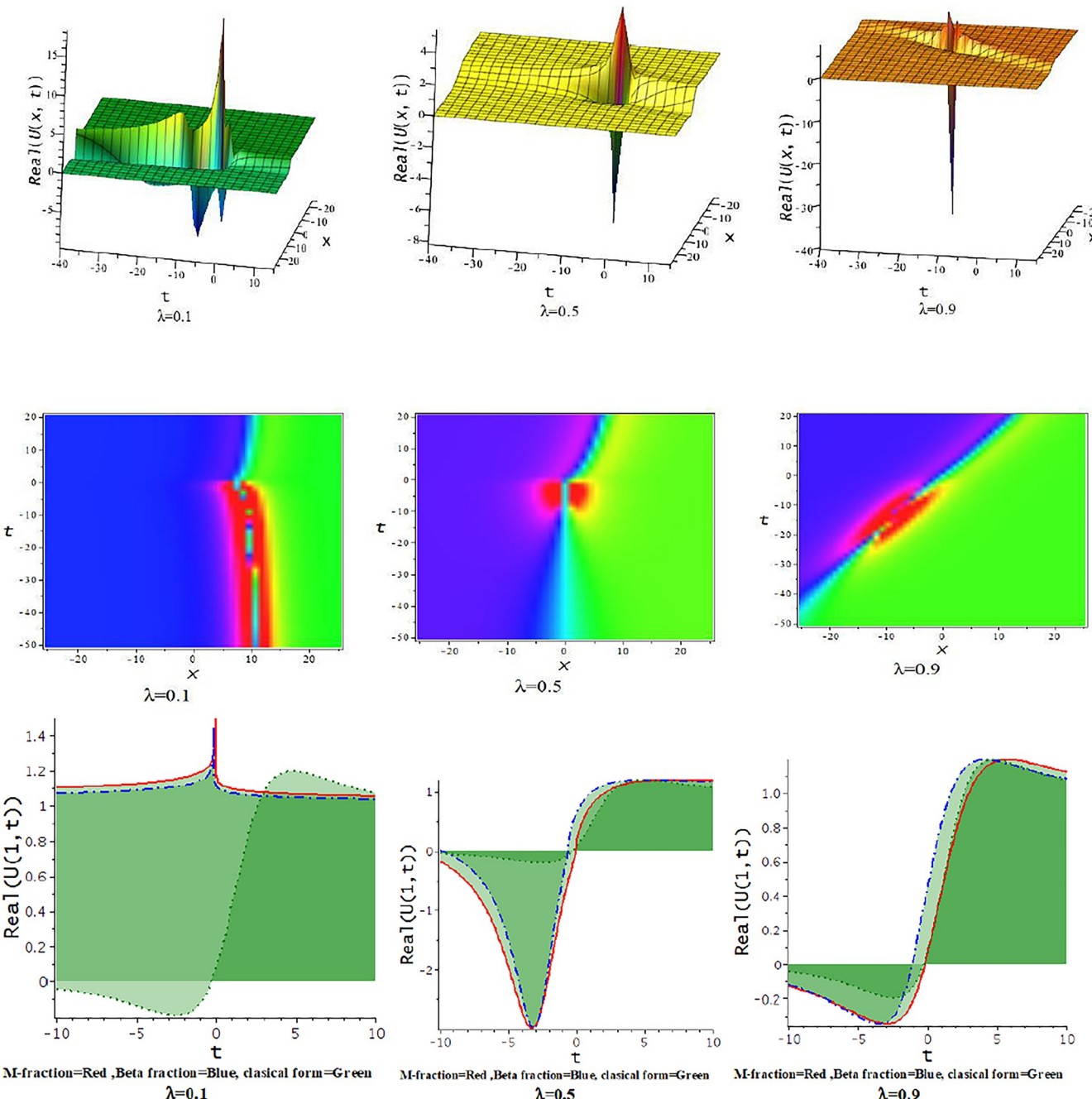

**Fig 2. The feature of collision of kink and lump wave of Eq (15) with change of the fractional parameter [$\lambda = 0.1, 0.5, 0.9$] at $d = 0.5 = \gamma, \alpha = 0.167, h = 5, \ell = -4, p = 1$.**

condensed matter physics. It explains how mass is created in some physical systems and how symmetry breaks spontaneously. The model holds special significance when considering the breaking of electroweak symmetry within the Standard Model of particle physics.

For $\gamma < 0$, this schema provides a hyperbolic function solution. The solutions, Eqs (30)–(33) and Eqs (38)–(41), are hyperbolic solutions. Periodic lump wave solution, Fig 11, illustrated via Eq (30) for the parameters $d = 0.5, h = -0.75, p = 1, l = 0.5, \gamma = -0.5, k = -0.25, a = 1, b = 0.5$.

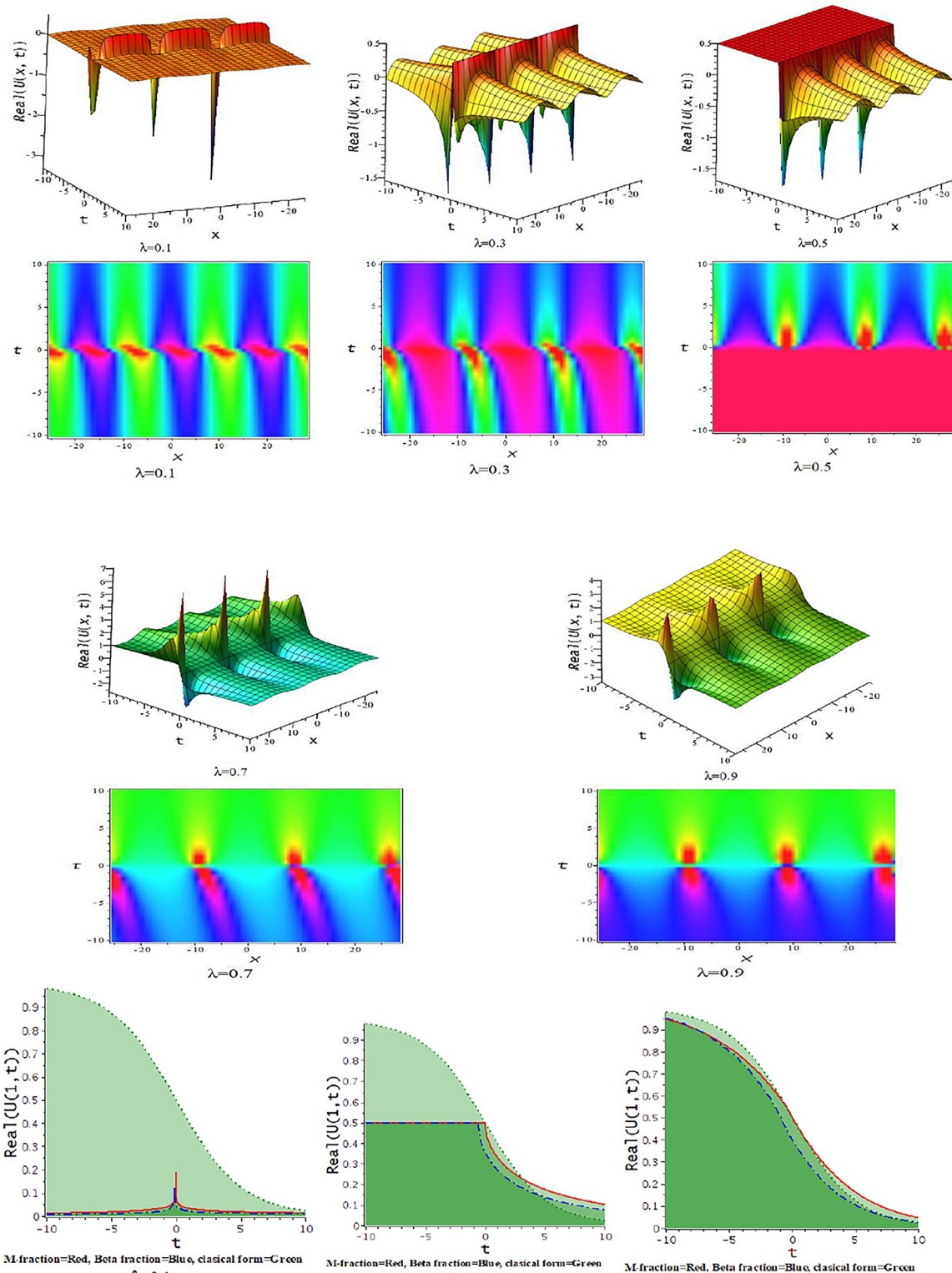

**Fig 3. The feature of lump wave solution of Eq (15) with change of the parameter as [λ = 0.1,03,0.5,0.7,0.9] at γ = 0.1,d = 0.5,h = 1, α = −0.25,ℓ = 0.5,p = 1.**

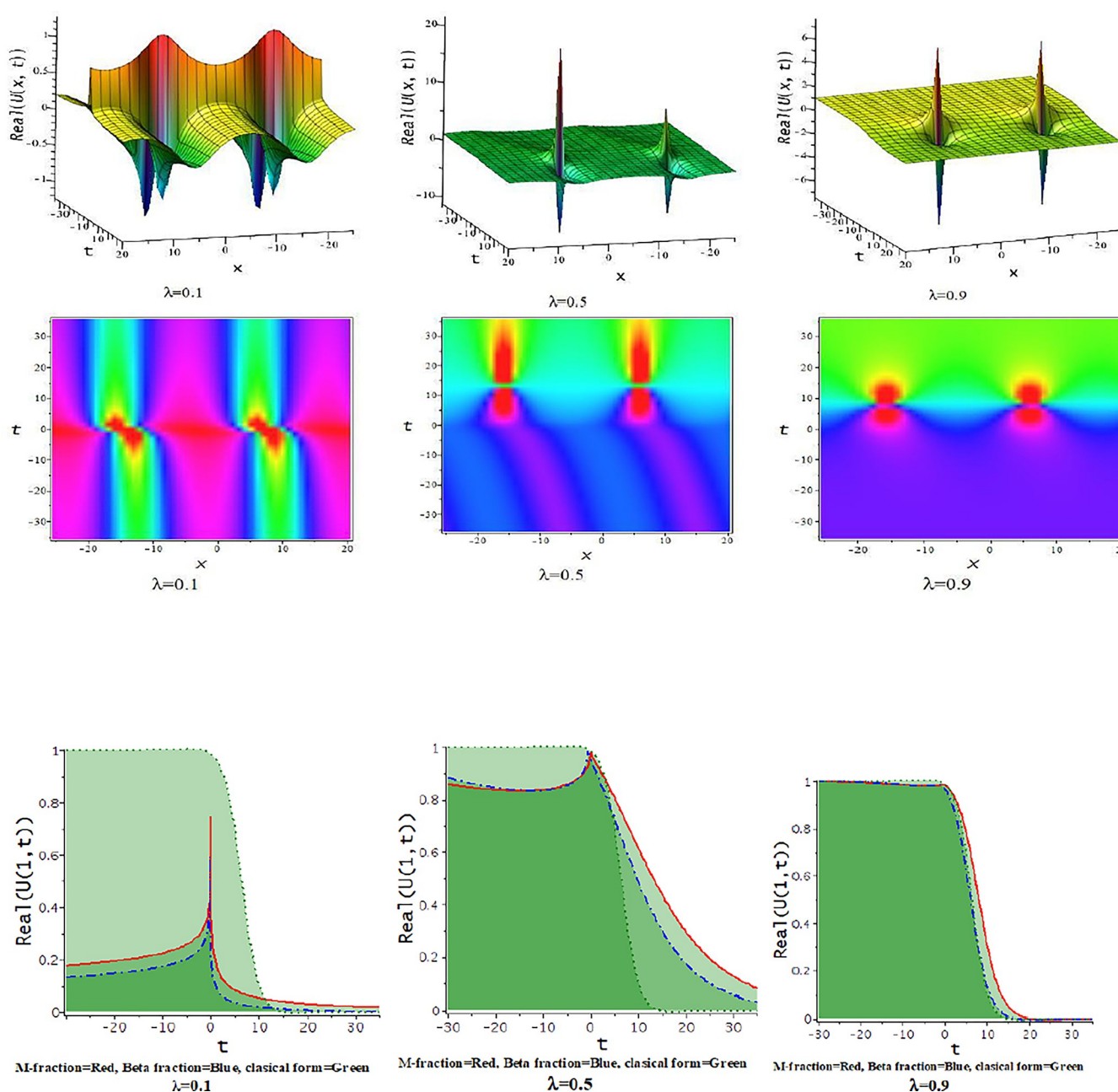

**Fig 4. The feature of lump wave solution of Eq (17) with change of the parameter as [λ = 0.1,0.5,0.9] at α = −0.167,d = 0.5,γ = 0.5,h = 5,ℓ = −4,p = 1.**

For $\gamma > 0$, the unified technique provides trigonometric function solutions, Eqs (34)–(37) and Eqs (42)–(45). The solutions, Eqs (31)–(33) represent lump wave solutions for the values involved constraints. The Eq (34) provides a periodic wave solution in Fig 12 for the parameters $d = 0.5, h = 1, p = 1, l = 0.5, \gamma = 0.20, k = 1, a = 0.1, b = −0.33$. The Fig 13 is a periodic solution of Eq (36) for the parameters $d = 0.5, h = 1, p = 1, l = 0.5, \gamma = 0.20, k = 1, a = 0.25, b = 0.33$.

In Fig 14, the feature of the soliton solution of Eq (38) with a change of the parameter $[\lambda = 0.3, 0.6, 0.9]$ at $p = 1, \ell = 0.5, d = 0.5, \gamma = −0.5, k = −1, h = −0.75, a = 0.5, b = 0.5$. In Fig 15, the feature of the kink soliton solution of Eq (40) with a change of the parameter

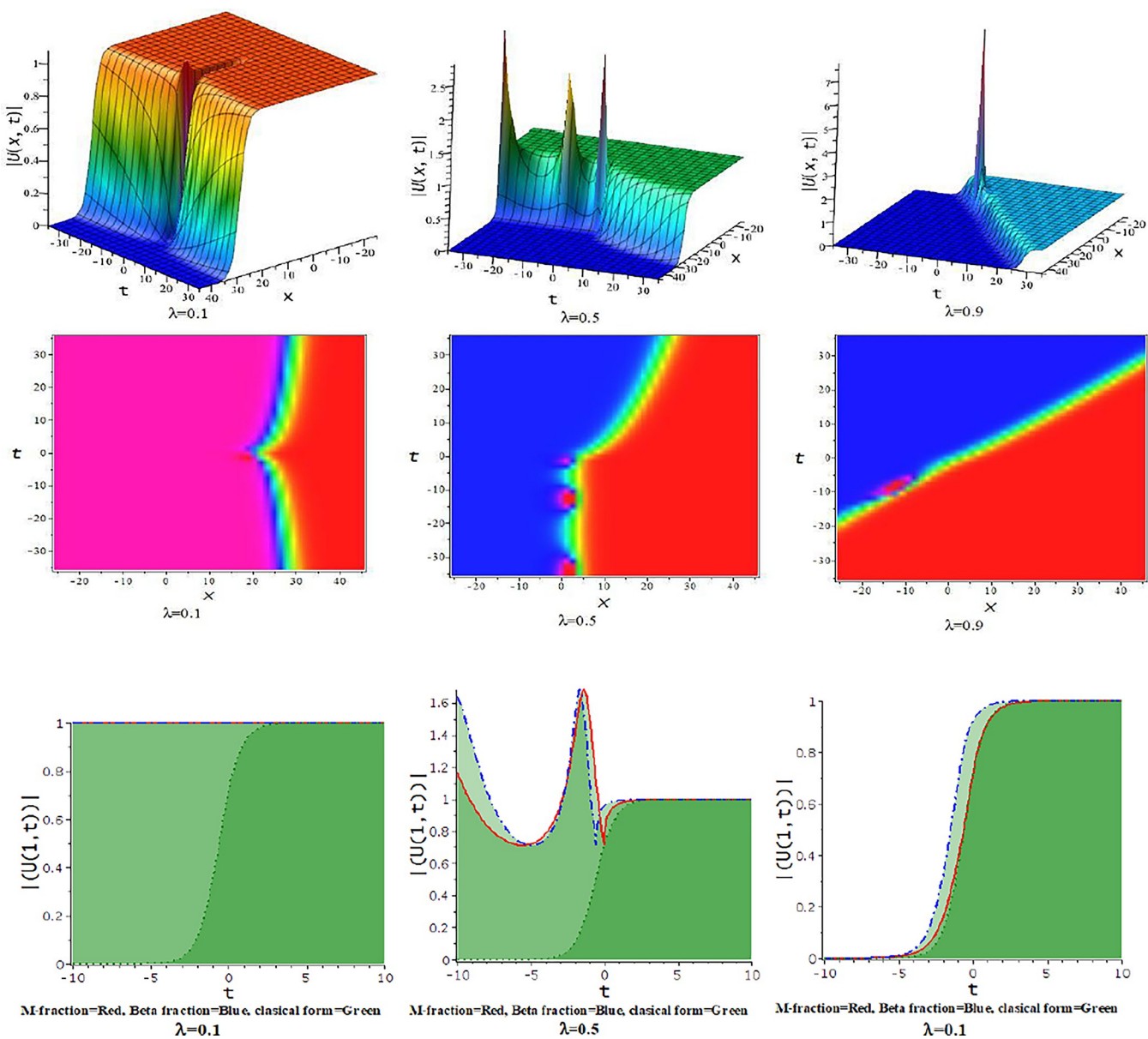

**Fig 5. The feature of collision of kink and lump wave of Eq (19) with change of the parameter as [λ = 0.1,0.5,0.9] at γ = −0.5, α = 1, h = −1, l = 0.2, d = 0.5, p = −1.**

$d = 1, h = 0.5, p = 1, l = 0.5, \gamma = -0.50, k = 2, a = 1.5, b = 1$. In Fig 16, the feature of the periodic soliton solution of Eq (45) with a change of the parameter
$p = 1, \ell = 0.5, d = 0.5, \gamma = 0.50, k = -1, h = 0.75, a = 0.5, b = 0.5$.

## 6. Comparisons and novelty of this manuscript

In this segment compares the attained solutions of Chafee-Infante equations with **Habiba et al** [39] and **Sakthivel et al** [37] and solution of Landau-Ginzburg-Higgs equations with **Barman et al** [44] and **Iftikhar et al** [46] solutions.

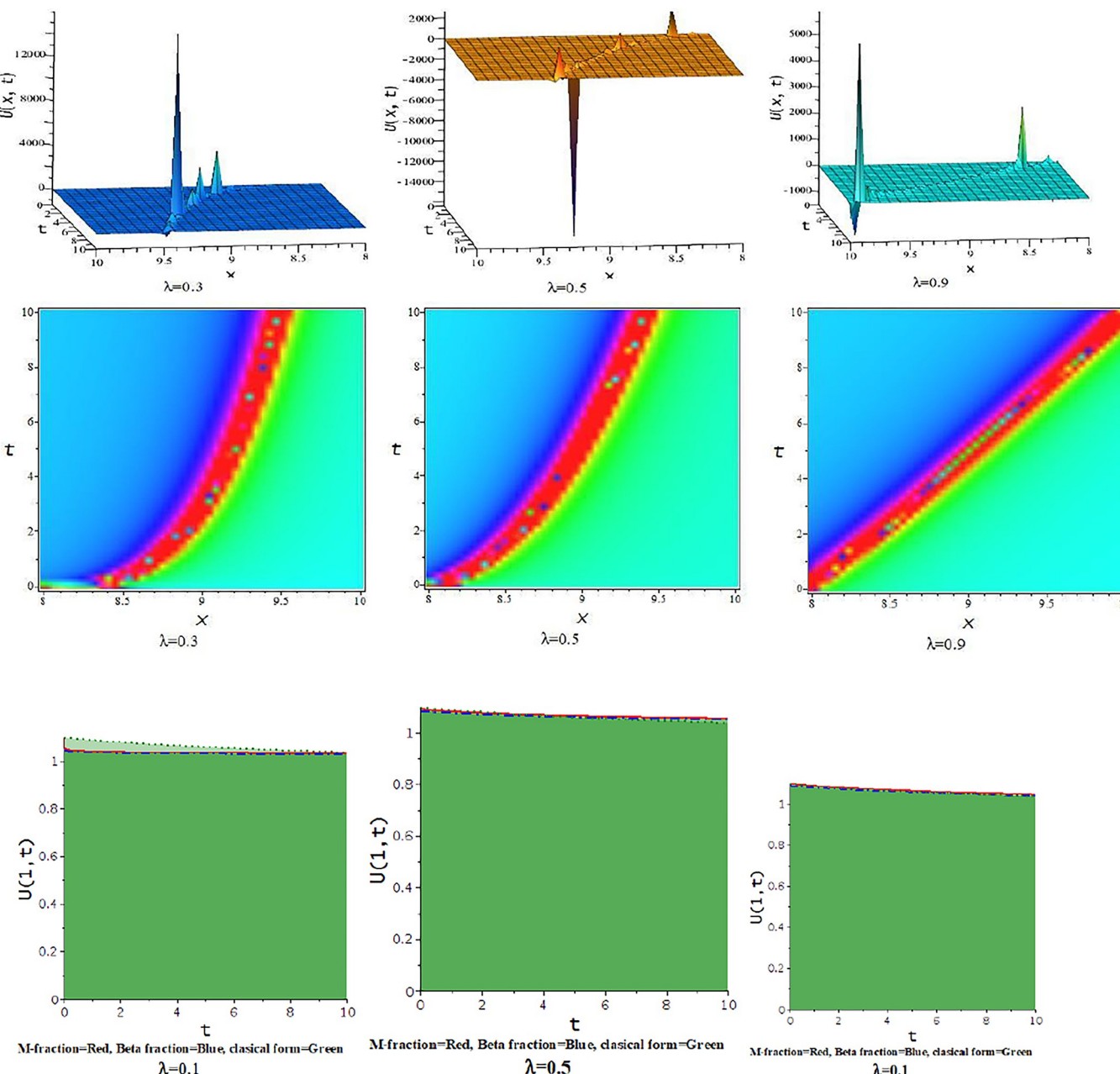

**Fig 6. The feature of soliton solution of Eq (19) with change of the parameter as [$\lambda = 0.3, 0.5, 0.9$] at $\gamma = -0.5, \alpha = 0.5, h = 0.25, d = 0.5, p = -1$.**

### Improved Kudryashov method and Exp function method for C-I equation

**Habiba et al.** [39] investigated the solitary waveform solutions of the C-I model by the improved Kudryashov technique and found only exponential function solutions. The obtained solutions are represented as kink shape and anti-kink shape solutions for the numerical form. **Sakthivel et al.** [37] discovered the soliton waveform solutions to the classical form of Eq (1) using the exp function technique. They found eight solutions (Please see ref. [37]). Otherwise, we have originated sixteen solutions to Eq (1) by operating the unified method in this article.

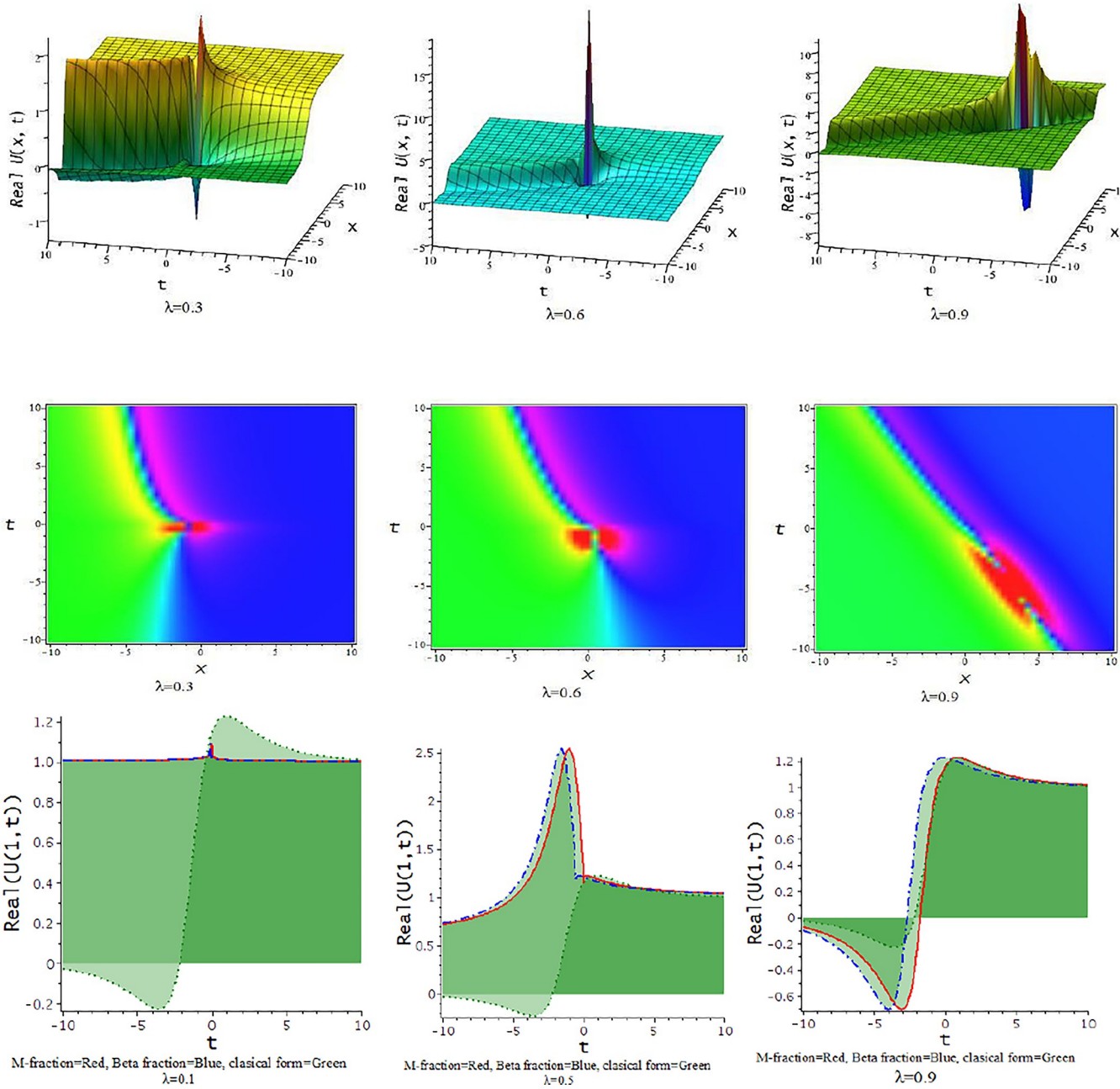

**Fig 7. The feature of collision of kink and lump wave of Eq ([23]) with change of the parameter as [λ = 0.3, 0.6, 0.9] at γ = 0.1, α = 0.5, h = 4, l = 0.5, d = 0.5, p = 1.**

### The extended tanh method and Two variable method for LGH equation

**Barman et al. [44]** explained some important waveforms of the LGH equation using the extended Tanh method and found only hyperbolic solutions. The attained solutions are embodied as bright-type and dark-type soliton, peakon-type, compact, and periodic solutions for the numerical form. **Iftikhar et al. [46]** discovered the soliton wave solutions to the classical form Eq ([2]) utilizing the two variable methods and found only two solutions (Please see

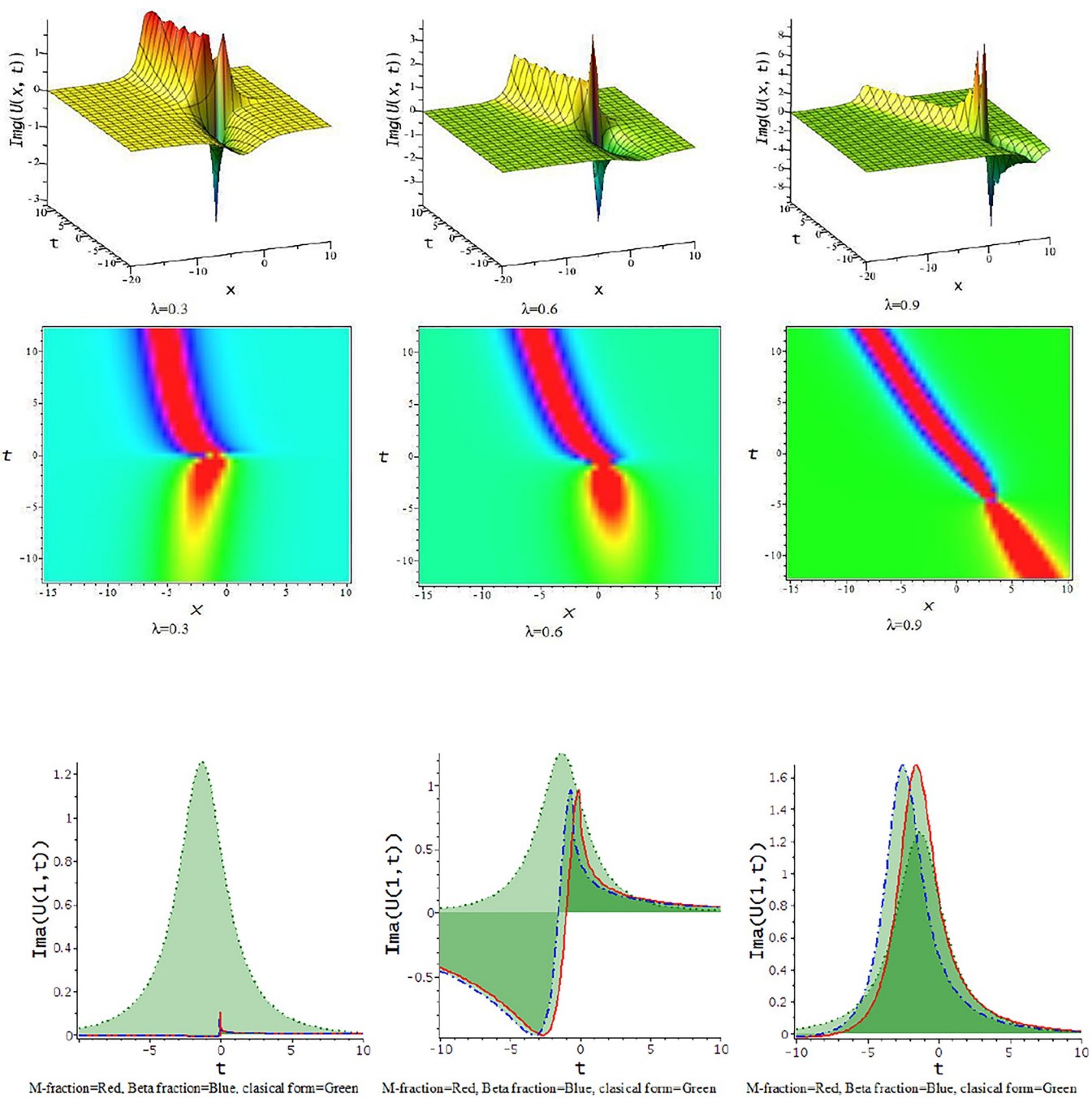

**Fig 8. The feature interaction of lump and bell type soliton solution of Eq (23) with change of the parameter as [λ = 0.3,0.6,0.9] at $\gamma = 0.1, \alpha = 0.5, h = 4, l = 0.5, d = 1, p = 1$.**

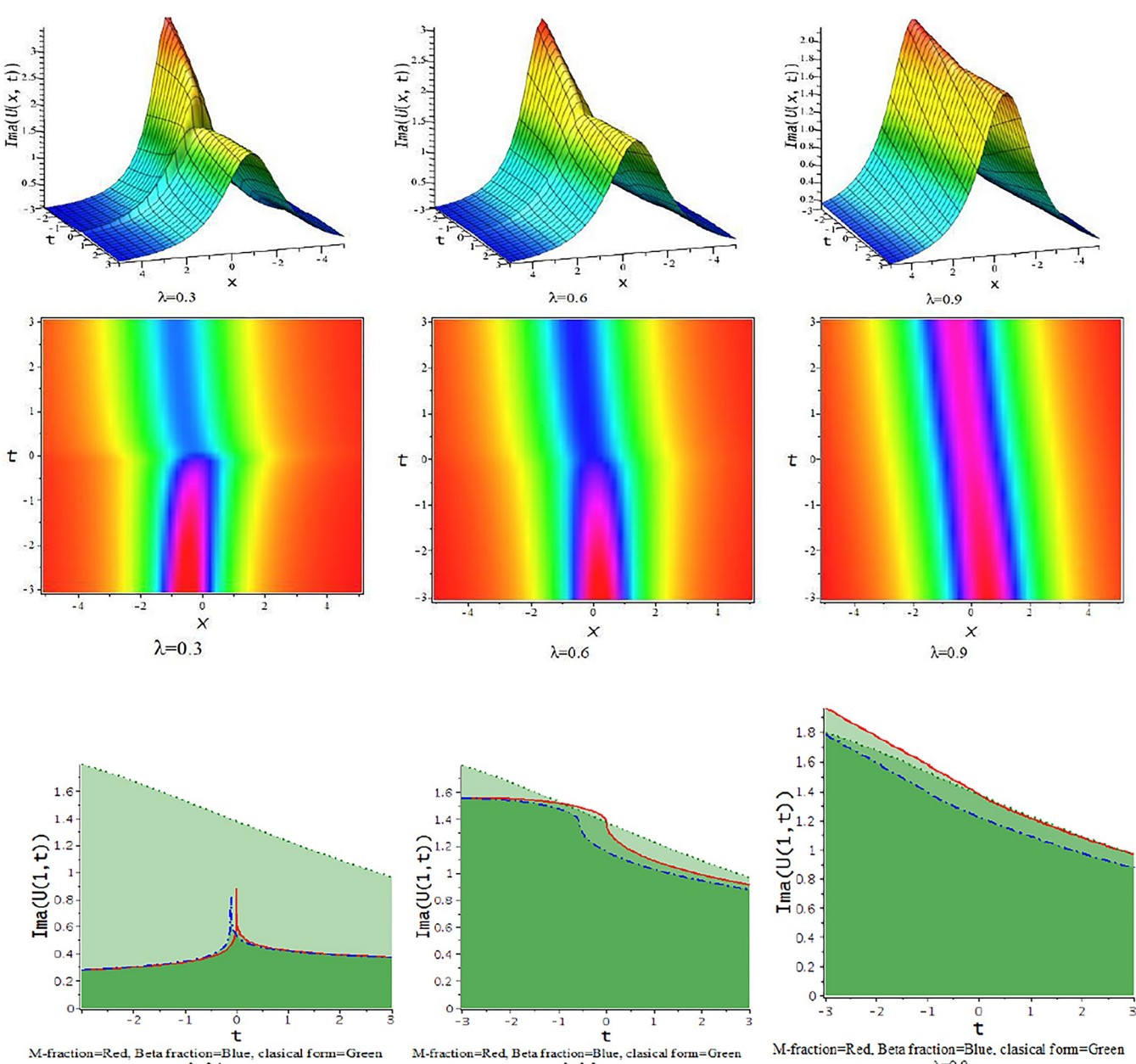

**Fig 9. The feature of bright bell shape solution of Eq (23) with change of the parameter as [λ = 0.3,0.6,0.9] at $\gamma = -0.5, \alpha = 0.5, h = 4, l = -4, d = 0.5, p = 1$.**

Ref. [44]). The obtained solutions are represented by two types of singular solutions in the numerical form. Otherwise, through employing the unified strategy in this article, we have discovered sixteen solutions to Eq (2). The Riccati equation is not the same for both methods.

Our Novelty:

In this paper, we utilized the unified technique to solve the C-I and LGH models. By using this method, we have found many solutions as trigonometric function, hyperbolic function, and rational function solutions. For the special values of the parameters we get kink shape, the collision of kink type and lump wave, the collision of lump and bell type, periodic lump wave, bell shape, and some periodic soliton waves for tM-fCI model and kink shape, periodic lump

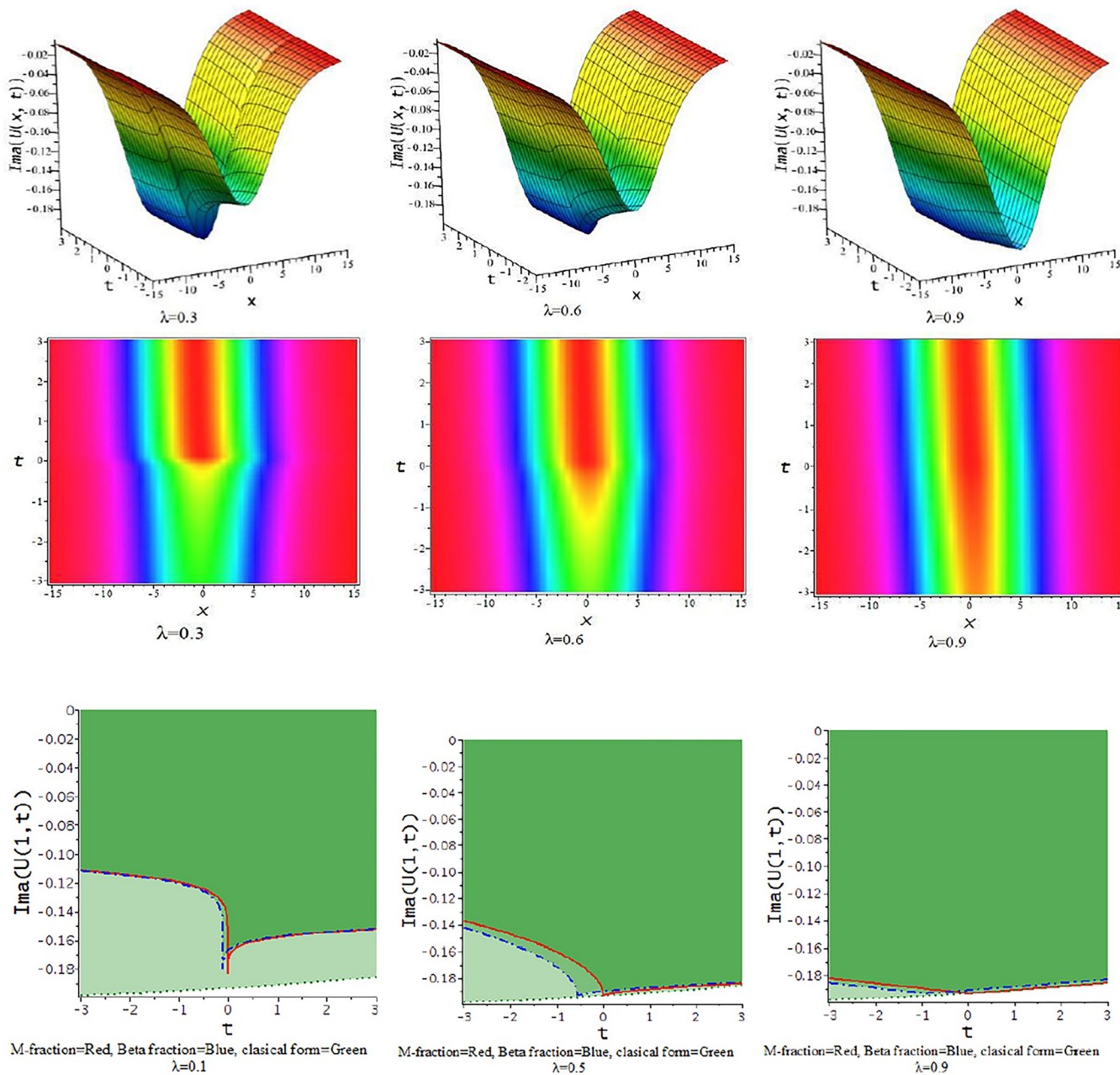

**Fig 10. The feature of dark bell shape solution of Eq (23) with change of the parameter as [λ = 0.3,0.6,0.9] at d = 0.5 = α = −γ, h = 5, l = −4, p = 1.**

wave, and some diverse periodic- and solitary-waves for tM-fLGH. From the above article, it is clearly that some of them are obtained first time for this model. Also at the first time we compared the effect of diverse fractional parameters on the obtained solutions.

## 7. Conclusions

In this article, more abundant new exact soliton solutions are successfully developed from two nonlinear truncated M-fractional models, the tM-fCI and tM-fLGH, by applying a unified scheme. The solutions are formed by rational, trigonometric, and hyperbolic functions under

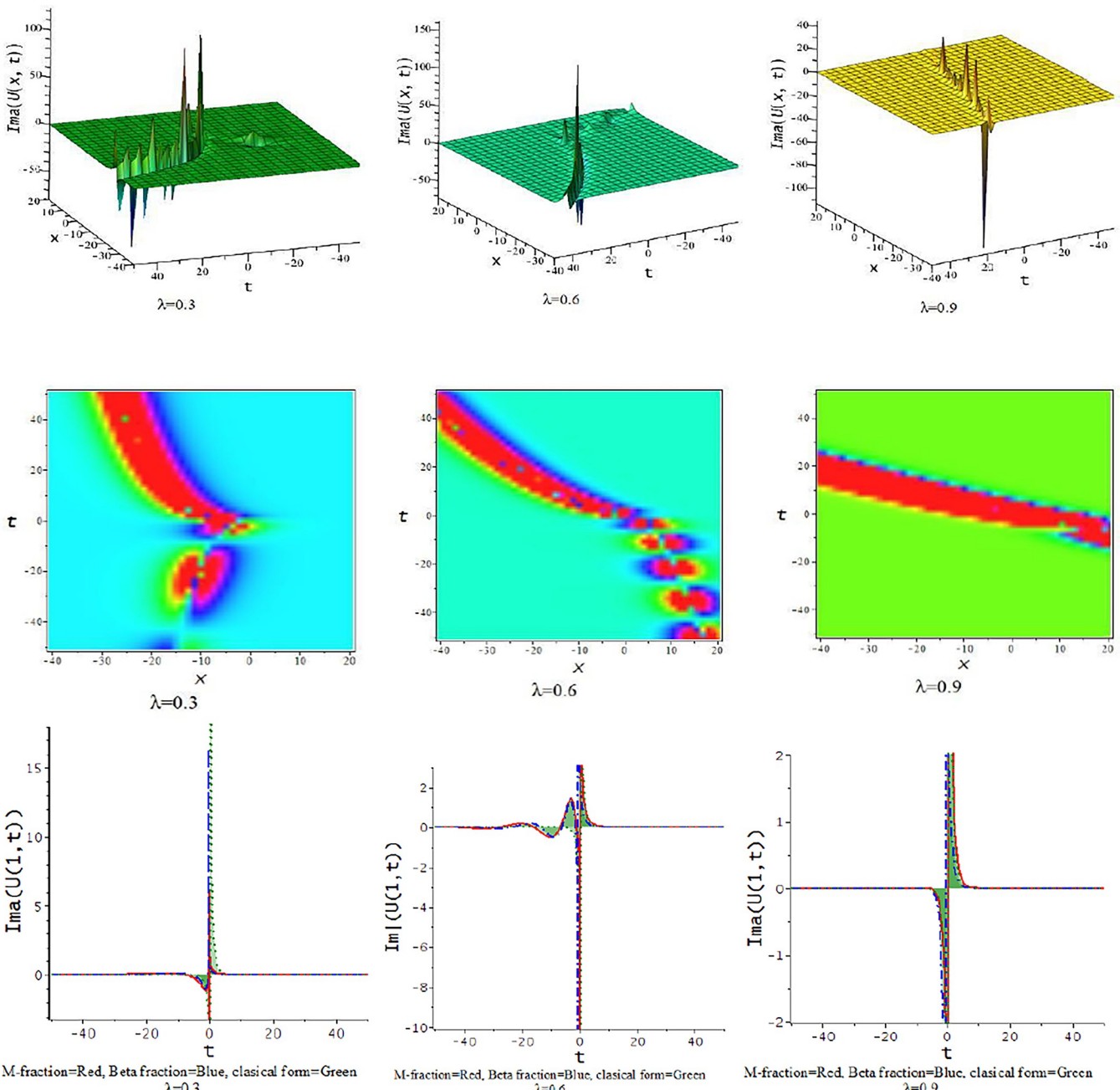

**Fig 11. The feature of periodic lump wave solution of Eq 30) with change of the constraints [λ = 0.3,0.6,0.9] at p = 1, l = 0.5, h = −0.75, γ = −0.5, k = −0.25, a = 1, d = 0.5, b = 0.5.**

the state of kink shape, the collision of kink type and lump wave, the collision of lump and bell type, periodic lump wave, bell shape, and some periodic soliton waves for tM-fCI and kink shape, periodic lump wave, and some diverse periodic- and solitary-waves for tM-fLGH successfully. Exactly, the amplitude and shape of the wave are reformed and changed due to slight changes in the fractional differential order. Moreover, effects of various fractional derivatives

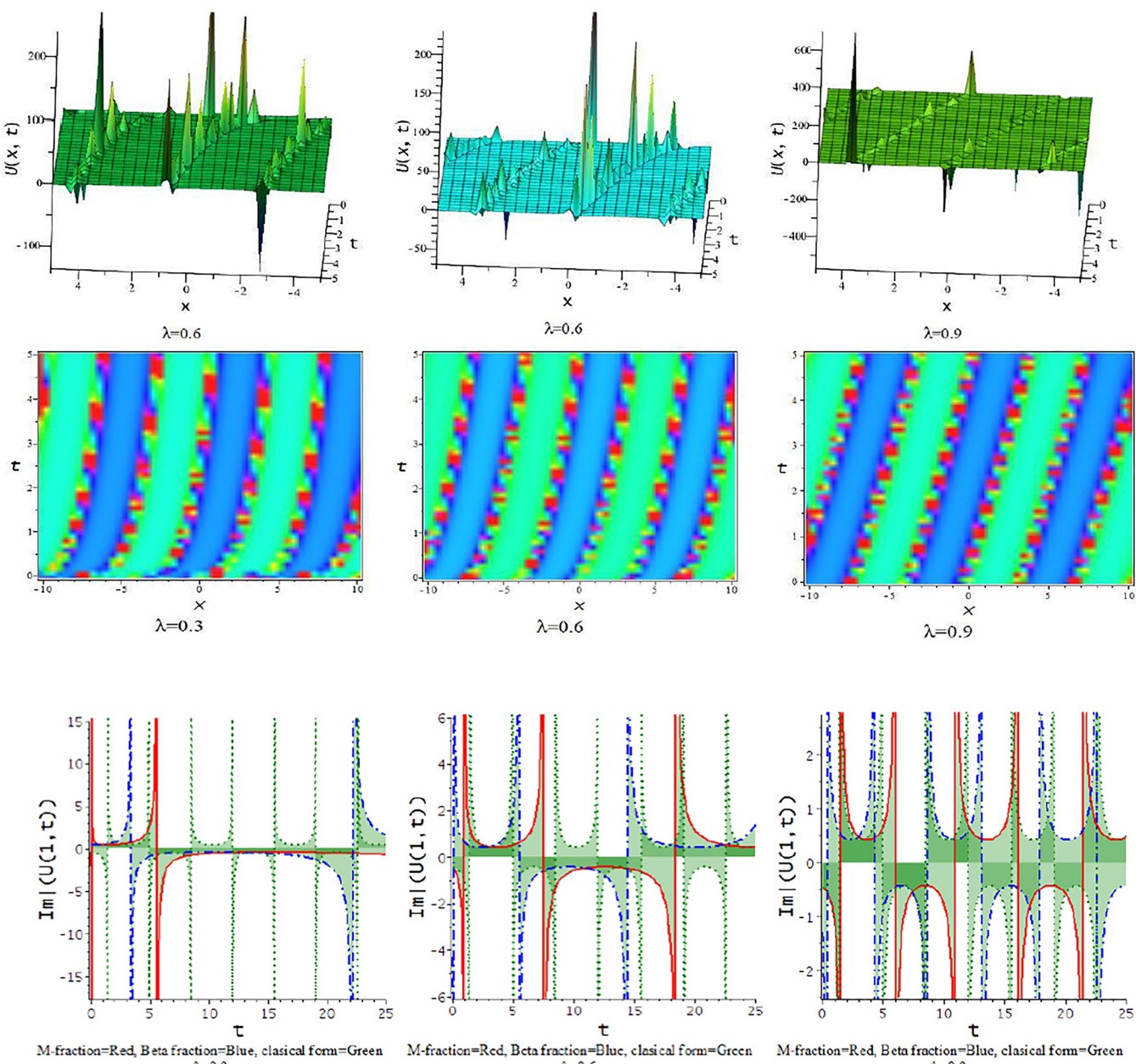

**Fig 12. The feature of periodic soliton solution of Eq (34) with change of the constraints [λ = 0.3,0.6,0.9] at**
$d = 0.5, h = 1, p = 1, l = 0.5, \gamma = 0.20, k = 1, a = 0.1, b = -0.33.$

are explored in the same 2-D graphics. All the solutions are illustrated with three-dimensional density plots. The output of our research shows that the projected method is an identical, effective, succinct, and strong mathematical tool for integrating complex nonlinear fractional models. In the future, spatio-temporal fractional derivation will be used for these models and also find some novel solitary wave solution by using generalized method.

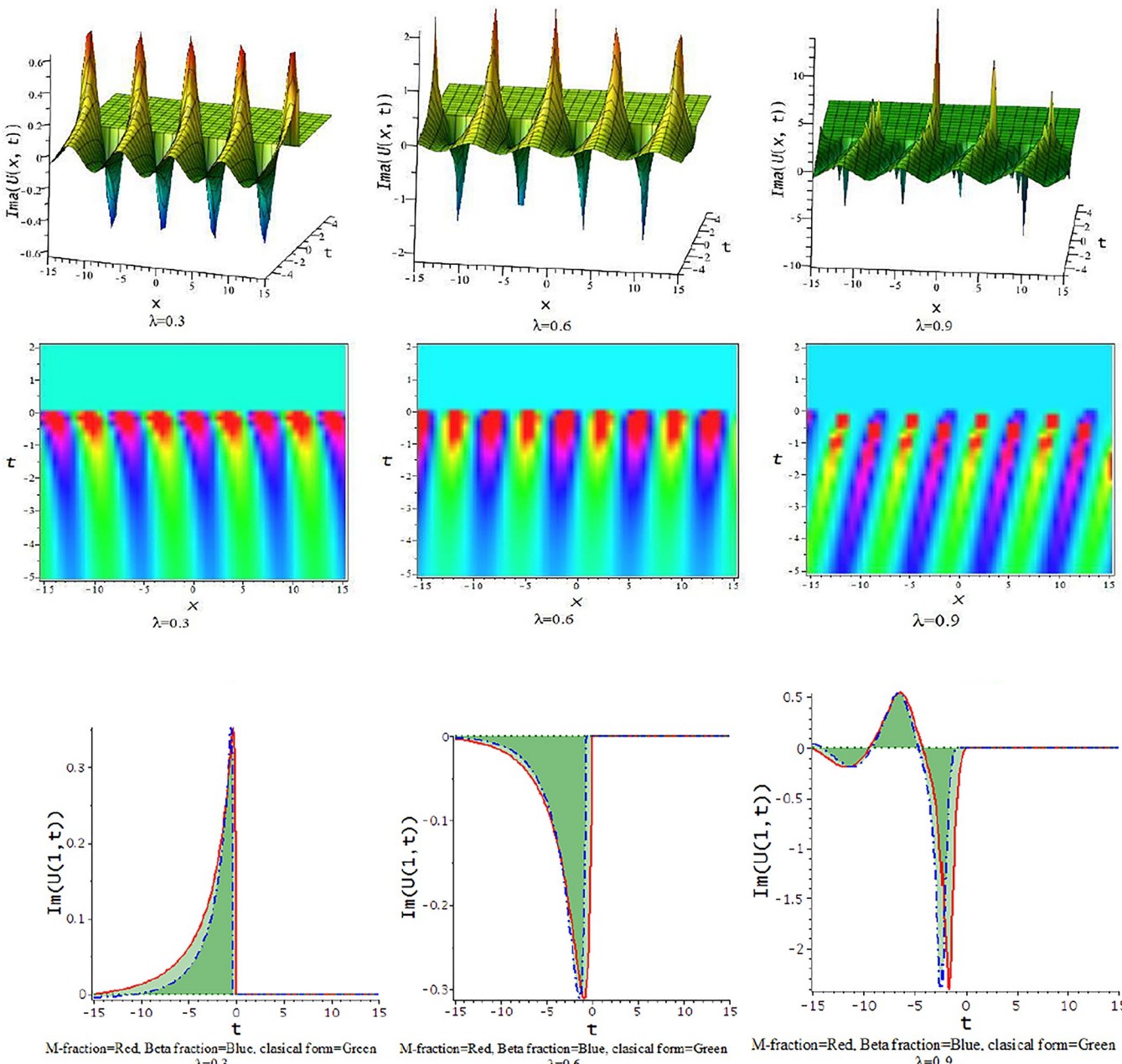

**Fig 13. The feature of periodic wave solution of Eq (36) with change of the constraints [λ = 0.3,0.6,0.9] at d = 0.5, h = 1, p = 1, l = 0.5, γ = 0.20, k = 1, a = 0.25, b = 0.33.**

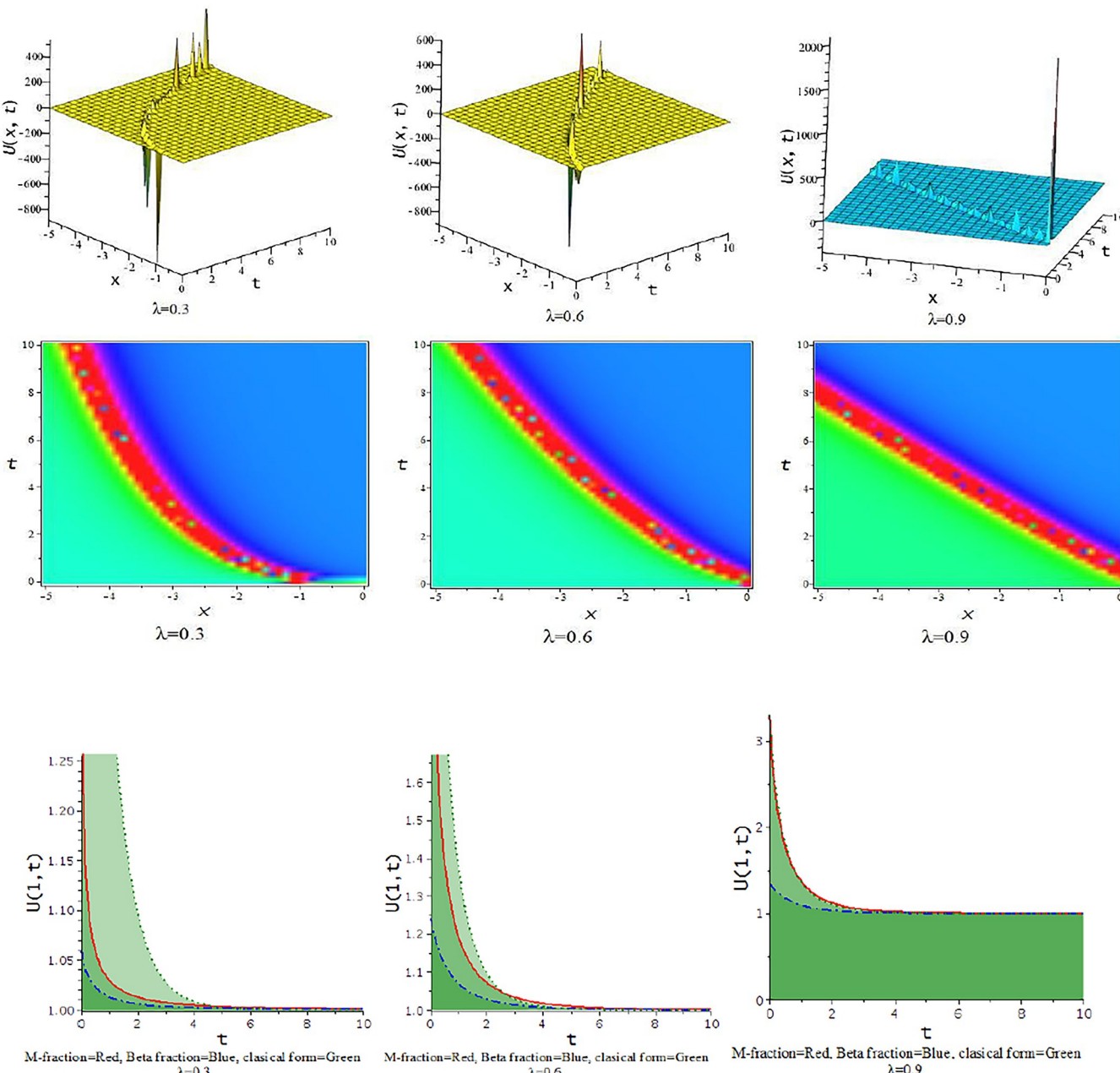

**Fig 14. The feature of soliton solution of Eq (38) with change of the constraints [λ = 0.3,0.6,0.9] at d = 0.5, h = −0.75, p = 1, ℓ = 0.5, γ = −0.5, k = −1, a = 0.5, b = 0.5.**

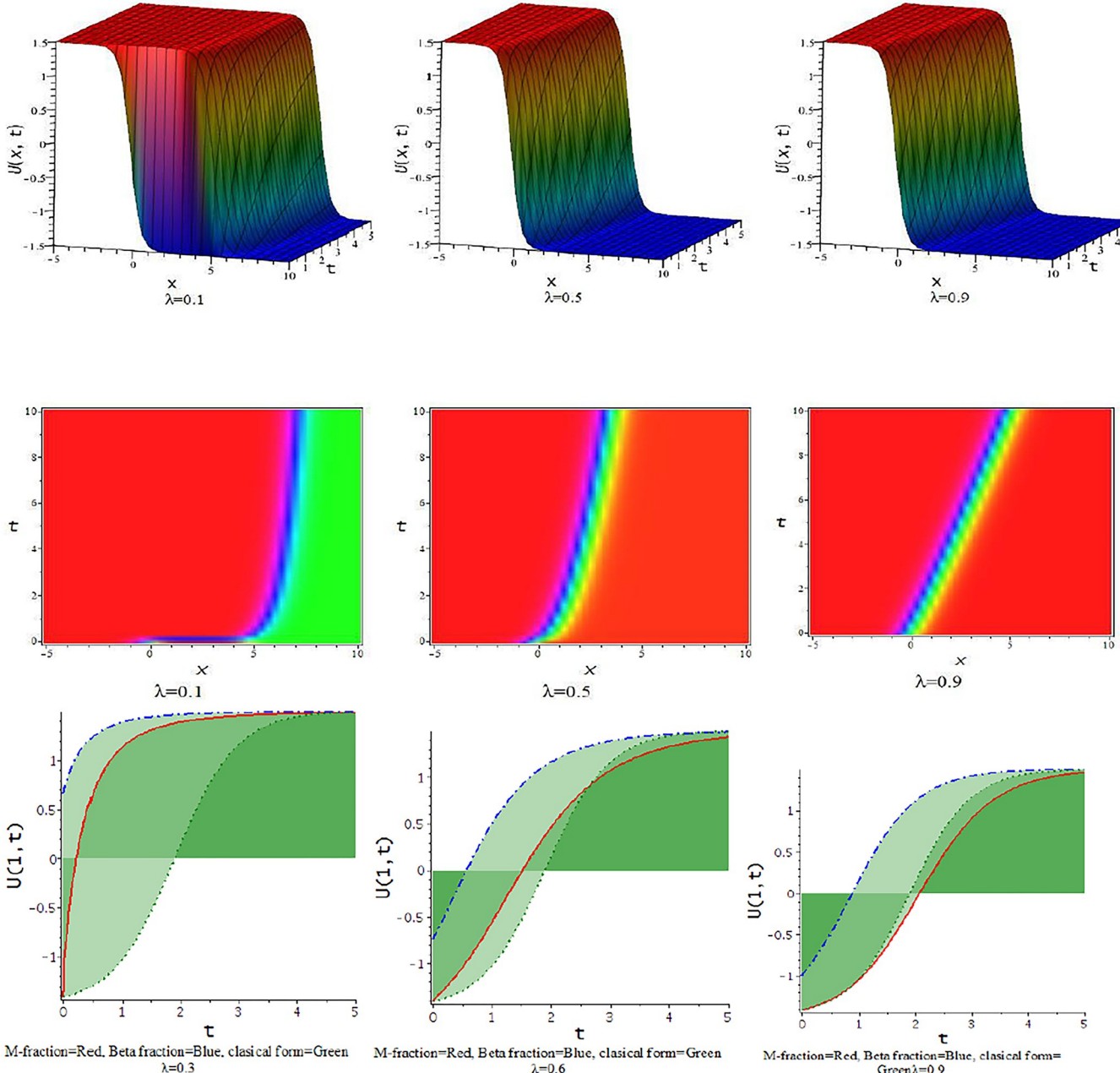

**Fig 15. The feature of kink soliton solution of Eq (40) with change of the constraints [λ = 0.3,0.6,0.9] at $d = 1, h = 0.5, p = 1, \ell = 0.5, \gamma = -0.50, k = 2, a = 1.5, b = 1.$**

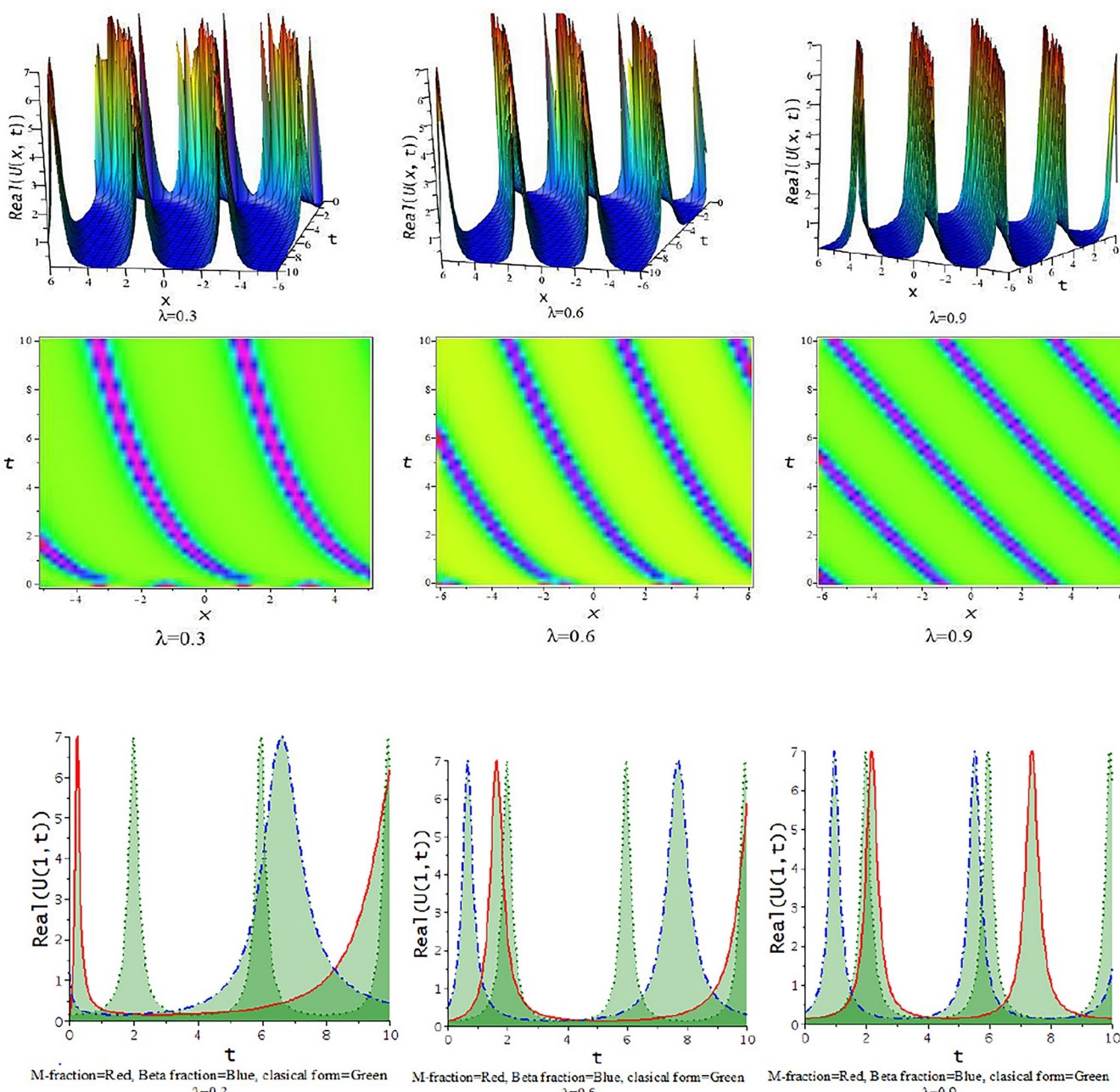

**Fig 16. The feature of periodic soliton solution of Eq (45) with change of the constraints [λ = 0.3,0.6,0.9] at d = 0.5, h = 0.75, p = 1, l = 0.5, γ = 0.50, k = −1, a = 0.5, b = 0.5.**

## Supporting information

**S1 File.**
(DOCX)

## Author Contributions

**Conceptualization:** Md. Mamunur Roshid, M. M. Rahman, Harun-Or Roshid, Md. Habibul Bashar.

**Data curation:** Md. Mamunur Roshid, M. M. Rahman, Harun-Or Roshid.

**Formal analysis:** Md. Mamunur Roshid.

**Investigation:** Harun-Or Roshid.

**Methodology:** Md. Mamunur Roshid, Md. Habibul Bashar.

**Software:** Md. Mamunur Roshid, M. M. Rahman, Harun-Or Roshid, Md. Habibul Bashar.

**Supervision:** Md. Mamunur Roshid.

**Validation:** M. M. Rahman.

**Writing – original draft:** Md. Mamunur Roshid.

**Writing – review & editing:** Md. Mamunur Roshid, Harun-Or Roshid.

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
