## [Decision Letter · Decision Letter 0]

14 Nov 2023

PONE-D-23-25341A Variety of soliton solutions of time M-fractional non-linear models via a unified techniquePLOS ONE

Dear Dr. Roshid,

Thank you for submitting your manuscript to PLOS ONE. After careful consideration, we feel that it has merit but does not fully meet PLOS ONE’s publication criteria as it currently stands. Therefore, we invite you to submit a revised version of the manuscript that addresses the points raised during the review process.

We look forward to receiving your revised manuscript.

Kind regards,

Ghulam Rasool

Academic Editor

PLOS ONE

“Include this sentence at the end of your statement: The funders had no role in study design, data collection and analysis, decision to publish, or preparation of the manuscript.”

Additional Editor Comments:

Revise.

Reviewers' comments:

Reviewer's Responses to Questions

**Comments to the Author**

1. Is the manuscript technically sound, and do the data support the conclusions?

Reviewer #1: Yes

Reviewer #2: Yes

2. Has the statistical analysis been performed appropriately and rigorously? 

Reviewer #1: N/A

Reviewer #2: Yes

3. Have the authors made all data underlying the findings in their manuscript fully available?

Reviewer #1: No

Reviewer #2: Yes

4. Is the manuscript presented in an intelligible fashion and written in standard English?

Reviewer #1: Yes

Reviewer #2: Yes

5. Review Comments to the Author

Reviewer #1: 1. Remove all typos from the text.

2. The introduction section should be improved by incorporating similar recent works.

3. Authors should highlight some applications of this study.

4. What is the novelty of this research work?

5. The explanation of obtained results should be backed by fundamental knowledge and principles

6. The English language should be well checked .

Reviewer #2: Each of the issues raised by the reviewers was addressed by the authors. The article was

revised based on the reviewers' comments and errors were corrected. I express my opinion that it is

appropriate for the article to be accepted for publication in present form.

6. PLOS authors have the option to publish the peer review history of their article (what does this mean?). If published, this will include your full peer review and any attached files.

Reviewer #1: No

Reviewer #2: No

---

## [Author Response · Author response to Decision Letter 0]

27 Jan 2024

Ans.: We done the necessary change to be the plos one journal style such as title, references, and figure number.

Ans.: we are agreed with the PLOS ONE the specific guidelines on code sharing process.

“Include this sentence at the end of your statement: The funders had no role in study design, data collection and analysis, decision to publish, or preparation of the manuscript.”

Ans.: we write your suggested statement at the end of the manuscript.

Ans.: I am sorry to inform you that we don’t used any data.

 Ans.: we added the ORCID iD for the corresponding author. 

Additional Editor Comments:

Revise.

Reviewers' comments:

Reviewer's Responses to Questions

Comments to the Author

1. Is the manuscript technically sound, and do the data support the conclusions?

Reviewer #1: Yes

Reviewer #2: Yes

2. Has the statistical analysis been performed appropriately and rigorously?

Reviewer #1: N/A

Reviewer #2: Yes

3. Have the authors made all data underlying the findings in their manuscript fully available?

Reviewer #1: No

Reviewer #2: Yes

4. Is the manuscript presented in an intelligible fashion and written in standard English?

Reviewer #1: Yes

Reviewer #2: Yes

5. Review Comments to the Author

Reviewer #1: 

1. Remove all typos from the text.

Ans.: Necessary correction has been done in the revised manuscript.

2. The introduction section should be improved by incorporating similar recent works.

Ans.: We added some recent article in introduction section. If you have any important and related article then please send me the doi number. 

3. Authors should highlight some applications of this study.

Ans.: We highlighted some applications of this study in section 1 and 5.

4. What is the novelty of this research work?

Ans.: In comparison section, we show the novelty of this article.

5. The explanation of obtained results should be backed by fundamental knowledge and

 Principles.

Ans.: The obtained solutions are explored with its importance in section 5.1 and 5.2.

6. The English language should be well checked.

Ans.: English language has been checked properly.

Reviewer #2: Each of the issues raised by the reviewers was addressed by the authors. The article was revised based on the reviewers' comments and errors were corrected. I express my opinion that it is appropriate for the article to be accepted for publication in present form.

Ans.: Thanks for your recommendation.

---

## [Decision Letter · Decision Letter 1]

27 Feb 2024

A Variety of soliton solutions of time M-fractional non-linear models via a unified technique

PONE-D-23-25341R1

Dear Dr. Roshid,

We’re pleased to inform you that your manuscript has been judged scientifically suitable for publication and will be formally accepted for publication once it meets all outstanding technical requirements.

Kind regards,

Ghulam Rasool

Academic Editor

PLOS ONE

Additional Editor Comments (optional):

Accept.

Reviewers' comments:

Reviewer's Responses to Questions

**Comments to the Author**

1. If the authors have adequately addressed your comments raised in a previous round of review and you feel that this manuscript is now acceptable for publication, you may indicate that here to bypass the “Comments to the Author” section, enter your conflict of interest statement in the “Confidential to Editor” section, and submit your "Accept" recommendation.

Reviewer #1: All comments have been addressed

Reviewer #3: All comments have been addressed

2. Is the manuscript technically sound, and do the data support the conclusions?

Reviewer #1: Yes

Reviewer #3: Yes

3. Has the statistical analysis been performed appropriately and rigorously? 

Reviewer #1: Yes

Reviewer #3: Yes

4. Have the authors made all data underlying the findings in their manuscript fully available?

Reviewer #1: Yes

Reviewer #3: Yes

5. Is the manuscript presented in an intelligible fashion and written in standard English?

Reviewer #1: Yes

Reviewer #3: (No Response)

6. Review Comments to the Author

Reviewer #1: Authors should carefully check and remove all typos from the text.

In general acceptable and the paper can be accepted for publication.

Reviewer #3: The paper is well revised and it has the novel contribution on fractional model. Therefore, it can be accepted for the publication in PLOS ONE.

7. PLOS authors have the option to publish the peer review history of their article (what does this mean?). If published, this will include your full peer review and any attached files.

Reviewer #1: No

Reviewer #3: No

---

## [Editor Report · Acceptance letter]

21 Mar 2024

PONE-D-23-25341R1 

PLOS ONE

Dear Dr. Roshid, 

I'm pleased to inform you that your manuscript has been deemed suitable for publication in PLOS ONE. Congratulations! Your manuscript is now being handed over to our production team.

Kind regards, 

on behalf of

Dr. Ghulam Rasool 

Academic Editor

PLOS ONE